# PDEInvBench:
# A Comprehensive Dataset and Design Space Exploration of Neural Networks for PDE Inverse Problems

**Divyam Goel**                                                              *divyam123@berkeley.edu*
*Department of Computer Science, UC Berkeley*

**Nithin Chalapathi**                                                        *nithinc@berkeley.edu*
*Department of Computer Science, UC Berkeley*

**Sanjeev Raja**                                                             *sanjeevr@berkeley.edu*
*Department of Computer Science, UC Berkeley*

**Aditi S. Krishnapriyan**                                                   *aditik1@berkeley.edu*
*Departments of Computer Science and Chemical Engineering*
*UC Berkeley; LBNL*
**Reviewed on OpenReview:** *https://openreview.net/forum?id=MSjhqRnNyZ*

## Abstract

Inverse problems in partial differential equations (PDEs) involve estimating the physical parameters of a system from observed spatiotemporal solution fields, a fundamental task in numerous scientific domains. Neural networks are well-suited for PDE parameter estimation due to their capability to model function-to-function space transformations. While existing benchmarks of machine learning methods for PDEs primarily focus on the forward problem — mapping physical parameters to solution fields—to our knowledge, there are no similar comprehensive studies and benchmark datasets on PDE inverse problems, i.e., mapping solution fields to underlying physical parameters. We fill this gap by introducing *PDEInvBench* , a comprehensive benchmark dataset consisting of numerical simulations for both time-dependent and time-independent PDEs across a wide range of physical behaviors and parameters. Our dataset includes evaluation splits that assess performance in both in-distribution and various out-of-distribution settings. Using our benchmark dataset, we comprehensively explore the design space of neural networks for PDE inverse problems along three key dimensions: (1) optimization procedures, analyzing the role of supervised, self-supervised, and test-time training objectives on performance, (2) problem representations, where we study the value of architectural choices with different inductive biases and various conditioning strategies, and (3) scaling, which we perform with respect to both model and data size. Our experiments reveal several practical insights: 1) neural networks perform best with a two-stage training procedure: initial supervision with PDE parameters followed by test-time fine-tuning using the PDE residual, 2) incorporating PDE derivatives as input features consistently improves accuracy, and 3) increasing the diversity of initial conditions in the training data yields greater performance gains than expanding the range of PDE parameters. We make our dataset and evaluation codebase freely available to facilitate reproducibility and further development of our work.

## 1 Introduction

Inverse problems involve inferring unknown parameters or governing laws of a physical system using observed data, such as measurements of system behavior over space and time. They are ubiquitous in numerous domains including geophysics (Tarantola, 2005), nanophotonics (Molesky et al., 2018), biomedical imaging (Vlaardingerbroek and Boer, 2013), and fluid dynamics (Karnakov et al., 2024). In the case of systems modeled by partial differential equations (PDEs), the inverse modeling task is to map observed spatiotemporal

solution fields to the underlying PDE parameters, which are typically physical constants such as viscosity, diffusivity, or the Reynolds number.

Recent advances in machine learning (ML) have shown promise in learning effective representations for PDE-related tasks (Rahman et al., 2023; Tran et al., 2023; Herde et al., 2024). In particular, neural operators (NO) (Li et al., 2021; Kovachki et al., 2023) have emerged as powerful tools for modeling the mapping between function spaces. While significant progress has been made in establishing effective NO recipes for the *forward* problem (Gupta and Brandstetter, 2022; Takamoto et al., 2022; Lu et al., 2022)—predicting spatiotemporal solution fields from known PDE parameters—there has been comparatively less investigation into PDE inverse problems. The ill-posedness of PDE inverse problems, including issues of non-existence, non-uniqueness, and instability with respect to noisy observational measurements (Engl et al., 1996; Tarantola, 2005; Kitanidis, 2010; Isakov, 2017), represents a challenging regime with distinct challenges compared to the forward problem.

In this work, we seek a principled understanding of the design space of neural network approaches — both within and outside the neural operator framework — for solving PDE inverse problems. As the first step, we introduce *PDEInvBench*, which is, to our knowledge, the first comprehensive dataset dedicated to benchmarking the performance of ML approaches for solving PDE inverse problems. Our dataset consists of high-quality numerical simulations of 5 PDE systems at various spatial resolutions, including Darcy flow (241x241, 421x421), reaction-diffusion (128x128, 512x512), unforced (64x64, 256x256) and forced Navier-Stokes (64x64, 2048x2048), and Korteweg-De Vries (256), at a wide range of physical parameter values and initial conditions. The PDEs cover a range of mathematical forms, including elliptic, hyperbolic and parabolic (Evans, 2022). Similarly, parameter ranges are chosen to exhibit a variety of behaviors including Turing bifurcations, diffusion, steady-states, turbulence, and laminar flow. We construct evaluation splits for each PDE system to investigate performance in both in-distribution and various challenging out-of-distribution settings.

Our second main contribution is the comprehensive benchmarking of ML-based methods for PDE inverse problems. We organize our investigation around three fundamental design axes:

1. **Model optimization procedure:** What is the optimal strategy for training neural networks for PDE inverse problems? Is a supervised loss from paired parameter-solution data sufficient, or is it also beneficial to incorporate physics-based PDE residual terms? How can test-time adaptation be used to improve generalization to new parameter regimes without explicit data?

2. **Problem representation and inductive bias:** The choice of neural architecture encodes implicit assumptions about the underlying problem. Do spectral biases (as in the Fourier Neural Operator), local convolutional patterns (as in ResNets), or global attention mechanisms (as in Transformers) best capture the relationship between solution fields and physical parameters? Should features like the derivative terms of the PDE be explicitly provided as input, or left for the network to learn implicitly?

3. **Scaling properties:** Resource constraints often force tradeoffs. Is it more beneficial to increase model size or to generate more training data? When generating data, should one prioritize covering more physical parameter values or more initial conditions?

We evaluate models using standard ML metrics such as relative error and the slope of scaling curves. We also perform physically motivated evaluations, such as checking whether the energy spectra of numerical simulations evolved with the predicted PDE parameter match that of reference simulations.

From our extensive experiments, we distill key insights that can guide practitioners in deploying neural networks for inverse problems.

- A two-stage training approach is preferable when learning neural networks for PDE inverse problems: 1) training with a supervised data loss, followed by 2) test-time training using the PDE residual.

- Explicitly conditioning networks on spatial and temporal derivatives significantly improves performance across all models and datasets.

- For time-dependent PDEs, the FNO architecture generally outperforms ResNet and Transformer-style architectures. However, given the vast space of architectural factors, more studies should be conducted to better understand how model inductive biases affect predictions for PDE inverse problems.

- For a fixed computational budget, generating multiple trajectories for each parameter value using different initial conditions yields greater performance gains than expanding parameter coverage.

We release the PDEInvBench dataset[1] and evaluation codebase[2] as a modular, standardized environment designed to facilitate systematic exploration of the PDE inverse problem design space and serve as a foundation for future methodological advancements by the community.

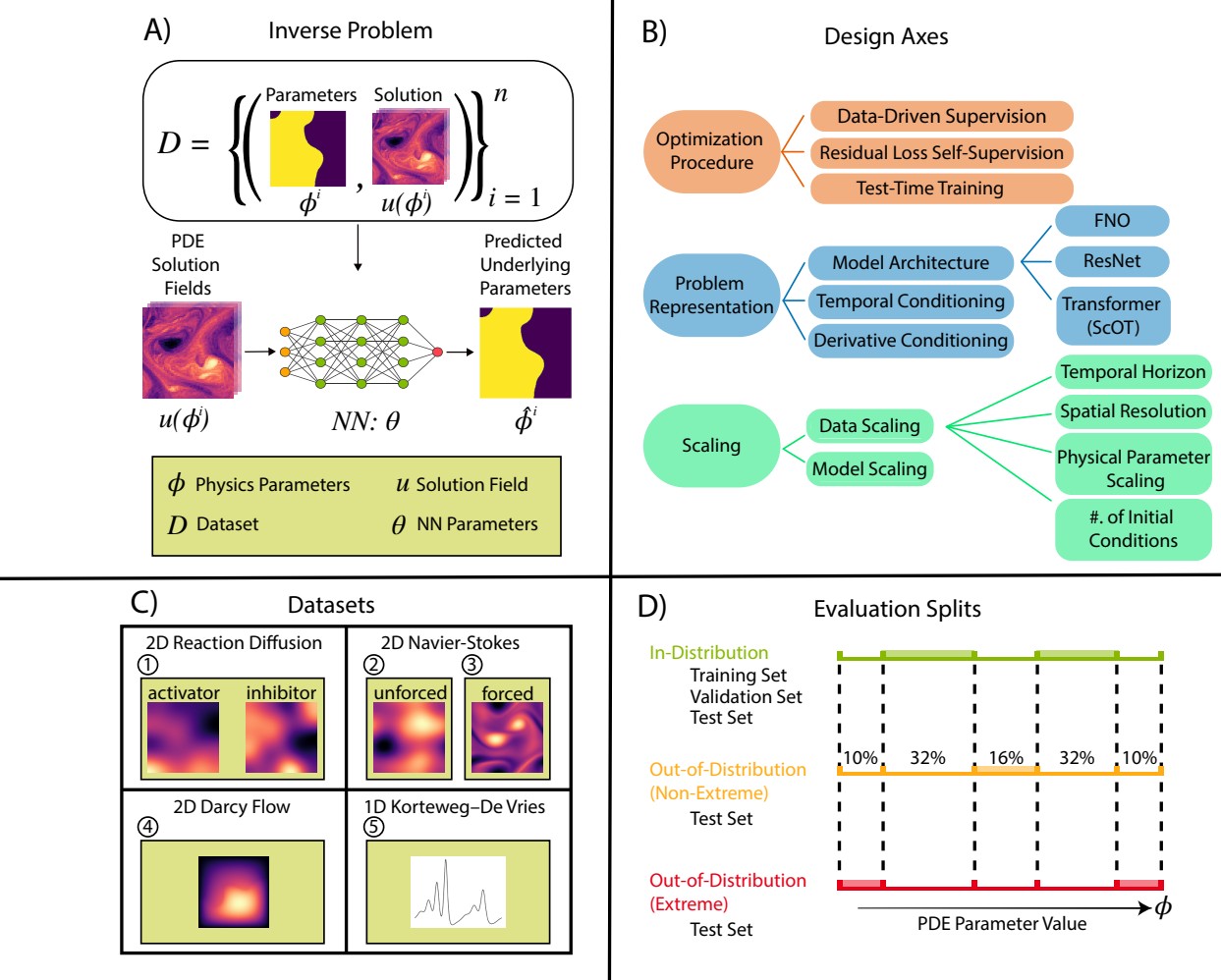

Figure 1: **Overview of design investigation of neural networks for PDE inverse problems.** (**A**) We consider the inverse problem setting in which a neural network learns to map from PDE solution fields $u(\phi)$ to predicted PDE parameters $\hat{\phi}$, using a dataset $D$ consisting of parameter-solution pairs as supervision. (**B**) Our investigation is split across three key design axes: optimization procedures, problem representation, and scaling properties. (**C**) We benchmark on diverse datasets spanning different PDE families, including 2D Reaction Diffusion, 2D Navier-Stokes, 2D Darcy Flow, and 1D Korteweg-De Vries. (**D**) We perform evaluations on both in-distribution parameter regimes and various out-of-distribution scenarios to assess generalization capabilities.

## 2 Related works

**Neural operators.** Neural Operators (NOs) are a class of discretization-invariant, universal function approximators which learn mappings between infinite-dimensional function spaces, making them particularly

---

[1] https://huggingface.co/datasets/DabbyOWL/PDE_Inverse_Problem_Benchmarking
[2] https://github.com/ASK-Berkeley/PDEInvBench

well-suited for modeling PDE problems (Kovachki et al., 2023). NOs typically involve a point-wise lifting operation, multiple layers of a learnable iterative kernel integration, followed by another point-wise projection to the output function space. For a mathematical description of the NO framework, see Appendix A. Several types of NOs have emerged with different inductive biases. The Fourier Neural Operator (FNO) (Li et al., 2021), including its many extensions (Tran et al., 2023; Kochkov et al., 2021) and generalizations (Du et al., 2024), leverages a spectral representation by parameterizing the integral kernel in the Fourier domain, enabling efficient modeling of global dependencies and smooth functions that naturally arise in physical systems. Transformers adapted for PDEs—like the scalable Operator Transformer (scOT) (Herde et al., 2024) which modifies the Swin-Transformer (Liu et al., 2022)—use self-attention to model both local and global dependencies in a manner compatible with the NO framework. Multiple other attention-based architectures have been proposed (Cao, 2021; Hao et al., 2023a; Alkin et al., 2024; Wu et al., 2024), but we use scOT, which is implemented in the HuggingFace Transformers library (Wolf et al., 2020).

**Inverse modeling using neural networks and Neural Operators.** Several studies have explored neural networks for inverse PDE parameter estimation. Li et al. (2021) use a function space Markov chain Monte Carlo method to recover the distribution of initial conditions using FNO as a surrogate numerical solver. Li et al. (2024) extend this approach with a physics-informed losses during training and test-time refinement (similar to the test-time training (TTT) method we benchmark), enabling both direct inverse modeling and gradient-based parameter estimation using forward surrogates. MacKinlay et al. (2021) investigate regularization terms when using gradient descent through a neural network forward surrogate to recover system parameters. Molinaro et al. (2023) propose Neural Inverse Operators (NIO), combining DeepONets with FNOs for recovering spatial coefficient fields from boundary measurements, and evaluate on electrical impedance tomography, inverse scattering, radiative transport, and seismic imaging datasets. While NIO proposes a specific architecture for a complementary set of PDE systems, our works focuses on benchmarking across architectures. Jiao et al. (2024) use DeepONets to identify diffusion coefficients on unknown manifolds. Latent Neural Operators (Wang and Wang, 2024) introduce a "Physics-Cross Attention" to map geometric inputs into a latent space for coupled forward-inverse prediction. Cho and Son (2025) provide theoretical stability guarantees for inverse problems and introduce PI-DION, an architecture tailored for solution field reconstruction and parameter estimation from partial measurements. Zheng et al. (2025) uses diffusion models to solve a broad class of inverse problems, but does not consider PDE-specific design choices, and does not consider multiple choices of PDE parameter. *PDEInvBench* advances this avenue of work by introducing a benchmark suite to accelerate model development and deriving relevant insights.

**Existing datasets and benchmarks.** Recent years have seen numerous PDE-specific benchmark datasets emerge, with most providing limited benchmarking of inverse problems, or consisting of numerical simulations performed with a limited range of PDE parameter values (Lu et al., 2022; Bhan et al., 2024; Toshev et al., 2024; Hao et al., 2023b). PDEBench (Takamoto et al., 2022) evaluates several PDE systems but only generate a few PDE parameters per system, while varying the initial conditions used to evolve solutions. PDEArena (Gupta and Brandstetter, 2022) examines Navier-Stokes across a range of bouyancy values, but focuses their analysis exclusively on the forward problem. BubbleML (Hassan et al., 2023), similar to PDEBench, only examines a limited number of parameters within the context of the forward problem. The Well (Ohana and others, 2024) is a large-scale dataset covering a broad set of PDEs with a wide range of physical parameter values. While their baselines and benchmarks target the forward problem, several datasets in the The Well (e.g., `acoustic_scattering`, `helmholtz_staircase`) are well-suited for inverse problems, and integrating them into our inverse problem evaluation framework is an important future direction. Kohl et al. (2023) focuses on fluid turbulence across a range of parameters, but benchmark autoregressive diffusion models only for the forward problem. By contrast, our dataset *PDEInvBench* is the only benchmark covering a wide range of PDE parameters (i.e., physical behaviors) and PDE systems in the inverse problem setting. Other works either exclusively focus on the PDE forward problem, or do not thoroughly study the inverse problem on multi-parameter datasets and across multiple design axes.

**System identification.** The PDE inverse problems we consider in this work are related to a broad line of literature on system identification Åström and Eykhoff (1971); Ljung (1998). Recent advances in multi-environment adaptation for system identification (Blanke and Lelarge, 2024; Kirchmeyer et al., 2022; Koupaï et al., 2024) focus on meta-learning task-agnostic representations with test-time adaptation of task-specific components according to the underlying parameters. However, these approaches primarily focus

on solving the forward problem and do not directly predict underlying parameters. In contrast, our focus is on directly predicting the underlying PDE parameters while leveraging knowledge of the governing PDE equations for test-time adaptation. This allows for straightforward test-time tuning without the need to distinguish between task-agnostic and task-specific representations.

## 3 Preliminaries

**Partial differential equations (PDEs).** We consider the PDEs of the form,

$$
\begin{aligned}
\mathcal{F}_\phi(u(x,t)) &= 0 & x &\in \mathbb{X}, t \in [0,T], \phi \in \Phi \\
\mathcal{B}(u(x,t)) &= 0 & x &\in \partial\mathbb{X} \\
u(x,0) &= u_0 & x &\in \mathbb{X}
\end{aligned}
\tag{1}
$$

$\mathcal{F}$ is a differential operator characterizing a family of PDEs, with solutions $u$ defined on the spatial domain $\mathbb{X}$ and temporal range $[0,T]$. $x,t$ define spatial-temporal points, and $\phi$ denotes physical parameters of the PDE family (e.g., diffusion coefficient, viscosity, density) drawn from a distribution of possible parameters $\Phi$. In general, $\phi$ may be an arbitrarily complex function of space and time, but in the systems we evaluate $\phi$ is typically a constant scalar or a spatially varying scalar field $\phi : \mathbb{X} \to \mathbb{R}$. $\mathcal{F}_\phi$ and $u$ are potentially highly non-linear in $x$ and $t$. $\mathcal{B}$ and $u_0$ denote boundary and initial conditions. For brevity, we abbreviate the entire solution over the spatial domain at time $t$ as $u_t$, i.e., $u_t = u(\cdot, t)$.

**Inverse problem and data loss.** The inverse problem denotes the task of learning a function $f_\theta$ with parameters $\theta$ that maps $k$ steps of observed PDE dynamics $u_{t-k}, ..., u_t$ from a PDE of the form of Equation 1 to the underlying PDE parameter $\phi$. Formally, the optimization problem is posed as,

$$
\theta^* = \underset{\theta}{\arg\min} \; \mathbb{E}_{\phi \sim \Phi} \| f_\theta(u_{t-k}, ..., u_t) - \phi \|_2,
\tag{2}
$$

where $u$ satisfies $\mathcal{F}_\phi(u) = \mathbf{0}$. In practice, the expectation is approximated by an average over a finite dataset $\mathcal{D}_{\text{train}} = \{\phi^i, u^i\}_{i=1}^N$ of $N$ PDE parameters paired with solution trajectories from a numerical simulation, where each $u^i$ satisfies $\mathcal{F}_\phi(u^i) = \mathbf{0}$. We refer to supervised learning using paired PDE parameter data as *data-driven supervision*. The data-driven inverse problem objective is then:

$$
\mathcal{L}_{\text{data}} = \frac{\| f_\theta(u_{t-k}^i, ..., u_t^i) - \phi^i \|_2}{\| \phi^i \|_2}.
\tag{3}
$$

Note that we use the relative error instead of the standard $L_2$ loss as physical parameters may span several orders of magnitude (e.g., viscosity $\nu \in [10^{-5}, 10^{-2}]$ in 2D turbulent flow). Our problem setting is challenging due to the presence of a range of PDE parameters $\Phi$, which induces a wide variety of physical behaviors in the solution $u$. For example, given two parameter settings $\phi_1, \phi_2 \in \Phi$, the solution to $\mathcal{F}_{\phi_1}(u(x,t)) = \mathbf{0}$ may be diffusive, while the solution to $\mathcal{F}_{\phi_2}(u(x,t)) = \mathbf{0}$ may converge to a nontrivial steady-state. This is distinct from prior works on neural PDE solvers which typically consider a single or a few values of $\phi$ and only vary the initial or boundary conditions. Additionally, PDE inverse problems may be ill-posed; distinct parameters $\phi_1 \neq \phi_2$ can produce observations that are close or indistinguishable, i.e., $|u^{\phi_1} - u^{\phi_2}|$ may be small even when $|\phi_1 - \phi_2|$ is large. The degree of ill-posedness depends on the number of observation frames, spatial resolution, and model input conditioning information. These factors are investigated in further depth in Section 6.2

**PDE residual loss.** We can incorporate physics-based constraints via the PDE residual $\| \mathcal{F}_\phi(u(x,t)) \|_2$, where the derivatives in $\mathcal{F}_\phi$ can be computed using autodifferentiation or a finite difference scheme. We elect to use the latter; henceforth, when $\mathcal{F}$ appears in a loss function, this refers to a finite difference approximation of the true differential operator on a finite-resolution mesh. We can use the PDE residual as a self-supervised learning signal Li et al. (2024). Given an inverse model $f_\theta$ that predicts parameters $\hat{\phi} = f_\theta(u_{t-k}^i, ..., u_t^i)$, the residual loss is,

$$
\mathcal{L}_{\text{res}} = \| \mathcal{F}_{\hat{\phi}}(u_{t-k}^i, ..., u_t^i) \|_2^2 = \| \mathcal{F}_{f_\theta(u_{t-k}^i, ..., u_t^i)}(u_{t-k}^i, ..., u_t^i) \|_2^2
\tag{4}
$$

**Test-time training.**  Test-time training (TTT) leverages the self-supervised nature of the residual loss for adaptation to specific parameters that may fall outside the training distribution. Given observed dynamics $u_{t-k},...,u_t$ at test time, we can adapt the inverse model $f_\theta$ via gradient updates on the following loss:

$$\mathcal{L}_{\text{Tailor}} = \mathcal{L}_{\text{res}} + \alpha \mathcal{L}_{\text{anchor}} \tag{5}$$

where $\alpha \in [0,1]$ is a weighting coefficient and,

$$\mathcal{L}_{\text{anchor}} = \frac{\|f_\theta(u_{t-k},...,u_t) - f_{\theta_{\text{frozen}}}(u_{t-k},...,u_t)\|_2}{\|f_{\theta_{\text{frozen}}}(u_{t-k},...,u_t)\|_2}, \tag{6}$$

where $\theta_{\text{frozen}}$ are model weights obtained after the initial training process. $\mathcal{L}_{\text{anchor}}$ prevents excessive deviations from the original model which we find helps stabilize tailoring.

## 4    Datasets and evaluation metrics

*PDEInvBench* contains five different PDE systems, each of which are simulated with a predefined range of PDE parameters chosen to span various physical behaviors including turbulence, steady-states, laminar flows, Turing patterns, and diffusion. A summary of the datasets are given in Table 1. For each sampled parameter value within the range (the number of distinct values and the sampling scheme is given in the "Number of Parameter Values" column), a fixed set of Gaussian random fields, serving as initial conditions, are evolved using a numerical solver according to the governing equations. After a fixed initial burn-in phase, solutions are recorded until reaching convergence (e.g., Turing bifurcations (Borckmans et al., 1995)) or for a preset time horizon. Details about the numerical solver parameters used, including the type of solver, time-stepping scheme, and burn-in periods, can be found in Appendix B.2 alongside a complete description of the governing equations in Appendix B.1. We also provide analysis of numerical convergence and energy spectra of the datasets in Appendix B.3 and Appendix B.4. Here, we provide a high-level description of all the datasets.

**2D Reaction Diffusion (RD) [parabolic].**  2D RD models two chemical species coupled non-linearly through the Fitzhugh-Nagumo equations (Klaasen and Troy, 1984; Takamoto et al., 2022). We focus on *activator-inhibitor* systems where the activator promotes the production of both species while the inhibitor suppresses both. There are 3 physical parameters $k$, $D_u$, and $D_v$. $D_u$ and $D_v$ are the diffusion coefficients for the activator and inhibitor, respectively. Meanwhile, $k$ represents the balance between both species or the threshold for excitement. Our dataset contains simulations with a range of values for $k$, $D_u$, and $D_v$, spanning a wide range of induced physical behaviors, from completely dissipative to Turing bifurcations (Borckmans et al., 1995). For simplicity, when training inverse models, we only learn over *one* parameter at a time (i.e., either $k$ or $D_u$) while treating the others as known. When $k$ is unknown, we refer to the dataset as 2D RD-$k$ and similarly, when $D_u$ is unknown as 2D RD-$D_u$.

**Unforced 2D Navier-Stokes (NS) [parabolic].**  The unforced NS equations govern fluid flow with no external energy injection (Du et al., 2024). The parameter range covers the laminar flow regime. Both the data and the solver use the vorticity form of the PDE, with a single parameter, viscosity $\nu$, serving as a proxy for internal friction.

**Forced 2D Navier-Stokes (TF) [parabolic].**  2D TF models fluids across a range of Reynolds numbers in the turbulent regime. We use a Kolmogorov forcing function term on the second wavenumber (i.e., $-2\cos(2y)$) (Kochkov et al., 2021). As with unforced NS, the physical parameter of interest for the inverse problem is the viscosity $\nu$.

**Korteweg-De Vries (KdV) [hyperbolic].**  KdV models waves on a shallow-water surface and includes a dispersive ($\partial_{xxx}u$) and an advection term ($u\partial_x u$) (Brandstetter et al., 2022). The physical parameter $\delta$ corresponds to the strength of the dispersive term (Zabusky and Kruskal, 1965).

**Darcy Flow (DF) [elliptic].**  Darcy Flow models the movement of a fluid through a porous medium (Li et al., 2021). All solutions converge to a non-trivial steady-state solution based on the diffusion coefficient field, making the system time-independent unlike previously described systems. The diffusion coefficient field, which is the parameter we learn over for the inverse problem, is a spatially varying scalar parameter rather than a constant scalar.

Table 1: Summary of PDE systems simulated in *PDEInvBench*. The dataset includes 4 time-dependent PDEs and 1 time-independent PDE, with physical parameter regimes for each chosen to span a wide range of physical behaviors, including turbulence, steady-states, laminar flows, Turing patterns, and diffusion.

| PDE System | Number of Partial Derivatives | Time-dependent | Spatial Resolutions | Parameter Range | Temporal Resolution (s)/ Time Horizon (s) | Number of Parameter Values | Number of ICs/Trajectories (per parameter combination) |
|---|---|---|---|---|---|---|---|
| 2D Reaction Diffusion (2D-RD) | 5 | ✓ | 128×128 | $k \in [0.005, 0.1]$, $D_u \in [0.01, 0.5]$, $D_v \in [0.01, 0.5]$ | [0.049 / 5] | $k$ - 2 linear-spaced, $D_u$ - 28 linear-spaced, $D_v$ - 27 linear-spaced | 5 |
| 2D Navier-Stokes Unforced (2D-NS) | 3 | ✓ | 64×64 | $\nu \in [10^{-4}, 10^{-2}]$ (Reynolds: 80-8000) | [0.0468 / 3] | 101 log-spaced | 192 |
| 2D Navier-Stokes Forced (2D-TF) | 3 | ✓ | 64×64 | $\nu \in [10^{-5}, 10^{-2}]$ | [0.23 / 14.75] | 120 log-spaced | 108 |
| Korteweg-De Vries (1D-KdV) | 4 | ✓ | 256 | $\delta \in [0.8, 5]$ | [0.73 / 102] | 100 linear-spaced | 100 |
| Darcy Flow (2D-DF) | 4 | × | 241×241 | Piecewise constant diffusion coefficient $a \in L^{\infty}((0,1)^2; \mathbb{R}^+)$ | - | 2048 (i.i.d Thresholded Gaussian) | - |

**Evaluation settings.** Each dataset consists of a broad range of parameter values for the training distribution. For all time-dependent PDEs, we create three different evaluation splits which test generalization in different settings. Splits are categorized in relation to the training parameter range. The *Out-of-Distribution (Extreme)* split, referred to as OOD Extreme, tests generalization to extreme parameter values. In the extreme set, parameters are drawn from the smallest and largest 10% of the parameter range. Similarly, the *Out-of-Distribution (Non-Extreme)* split, referred to as OOD Non-Extreme, tests generalization to out-of-distribution parameter values in the middle 16% of parameter values which are omitted from the training parameter range. Finally, the *In-Distribution* (ID) split consists of held-out parameters drawn from the training range. Figure 1(D) visualizes how splits are partitioned for the single parameter case. For systems with more than one PDE parameter (e.g., 2D Reaction Diffusion), we generalize the percentages to a hypercube with each axis representing a single parameter (see Appendix B.5) for a visual depiction.

For Darcy Flow (the only time-independent PDE in our dataset), the physical parameter is a spatial *field* which takes on one of two values, 3 or 12, in each position. We make only ID and OOD Extreme splits for Darcy flow. We define the splits by computing the fraction of grid points in each coefficient field that take on the maximum value (12), a statistic which is approximately normally distributed across the dataset. We partition coefficient fields according to this distribution using ±1.5 standard deviations, with the central mass defining the ID test set and the tails defining the OOD Extreme split (see Appendix B.5 for additional details).

**Evaluation metrics.** We present results using two primary metrics: *relative error* (↓) and *the negative slope of the line of best fit (NLS)* (↑). Relative error is defined as $||\phi - \hat{\phi}||_2 / ||\phi||_2$ where $\hat{\phi}$ and $\phi$ denote the predicted and true physical parameter, respectively. This metric is used to assess the quality of an inverse model. For experiments where we compare scaling approaches (e.g., relative error performance with respect to scaling the number of initial conditions versus scaling the number of generated physical parameters in Figure 4D-F), we use the NLS. From a plot of relative error versus a particular scaling axis, we compute a line of best fit, and we extract the NLS as the negative slope of that line. NLS measures the effect of scaling on the relative error by capturing the *decrease* in the $y$-axis (i.e., relative error) per $x$-axis unit (i.e., method of scaling). More negative values of the NLS indicate better scaling (i.e., faster reduction of the relative error due to scaling). This metric is related to well-established techniques used in the literature on neural scaling laws Hestness et al. (2017); Bahri et al. (2024) and machine learning force fields. We select the best performing models on the validation splits using these metrics (see Appendix B.6). In Appendix D.4, we present an additional evaluation to test the physical consistency of the predicted PDE parameters. We roll out numerical simulations with predicted PDE parameters and compare the resulting energy spectra with that of reference simulations run with the true PDE parameters.

# 5 Selected investigations for PDE inverse problems

We now describe our methodology for exploring key design axes for solving PDE inverse problems using neural networks. We investigate 1) optimization procedure, 2) problem representation and inductive biases, and 3) scaling properties, with several experiments within each axis. For all experiments, we train models with 3

different random seeds and report error bars of the corresponding standard deviation. In this section, we select the most significant investigations, but include additional experiments and details in Appendices C and D.

## 5.1 Optimization procedure

We investigate how different optimization approaches affect performance. Our exploration focuses on two key aspects: loss formulations and test-time training strategies.

**Loss functions.**   We consider two primary loss formulations: a supervised, purely data-driven setting with the data loss defined in Equation 3, and a purely self-supervised, residual-driven setting with the PDE residual loss defined in Equation 4. In Appendix D, we also consider a generalized "physics-informed" loss which combines the data-driven and PDE-residual setting.

**Test-time training.**   As defined in Section 3, we adapt a pretrained inverse model at inference time using a self-supervised objective (Equation 5). We take 50 gradient steps with a batch size of 32. We provide further details on test-time training in Appendix C.2

## 5.2 Problem representation and inductive biases

The choice of input features and inductive biases can significantly impact downstream performance. We investigate this through a comparative analysis of architectures and input representations.

**Architecture comparison.**   We compare four architectures in a parameter-controlled ($\sim 5$ million learnable parameters) manner (hyperparameters in Appendix C.3). The *Fourier Neural Operator (FNO)* (Li et al., 2021) serves as our baseline architecture and uses a spectral representation, similar to classical pseudo-spectral numerical methods. *ResNet* (He et al., 2016) tests the usefulness of locality in the spatial domain. *DeepONet* (Lu et al., 2021) tests the usefulness of separating the encoding of input fields from the representation of output coordinates via branch-trunk decomposition. Finally, we use the *scalable Operator Transformer (scOT)* (Herde et al., 2024), based on the Swin Transformers (Liu et al., 2022), to test the effectiveness of modeling local and global spatial dependencies through hierarchical attention mechanisms. Both FNO and scOT are neural operator architectures whose encoders are discretization invariant, and we include both to examine different classes of inductive biases developed for PDE applications. DeepONet is also a neural operator architecture, but differs in that it parameterizes the operator through a learned basis expansion rather than enforcing discretization invariance through convolutional or attention-based encoders. While not strictly a neural operator, ResNet operates over arbitrarily sized inputs due to the nature of convolutions which makes it a useful point of comparison. Note that the full inverse-problem pipeline (encoder followed by convolutional downsampling and an MLP regression head) is not discretization invariant for any architecture, as reducing a spatial field to a scalar parameter inherently requires resolution-dependent pooling and aggregation operations.

**Input representations.**   We examine the impact of explicitly providing derivative information as conditioning to the inverse model. In addition to the past $k$ observed frames $u_{t-k},...,u_t$, the partial derivatives appearing in the differential operator $\mathcal{F}$ are concatenated as additional channels to the input. Additionally, in Appendix D.2, we study the effect of varying the number of temporal conditioning frames $k$ provided to the inverse model.

**Partial observability.**   We include a brief study of partial observability in the inverse problem setting. We instantiate partial observability in two primary ways: 1) by introducing noise into the input solution fields (Appendix D.5), and 2) removing spatial grid lines to induce a non-uniform discretization (AppendixD.6).

## 5.3 Scaling properties

We conduct data scaling experiments along the number of initial conditions, total number of parameters and training time horizon, and model scaling experiments along the channel width (Appendix C.4).

**Data scaling.**   For each data scaling experiment, we evaluate the performance of FNO, ResNet, and scOT to isolate the effects of data quantity. We vary dataset size along three dimensions. In our *Initial Condition Scaling* experiments, we train on a variable number of unique initial conditions per PDE parameter value, namely 20%, 35%, 50%, 75%, and 100% of the total number of initial conditions. Similarly, in our *Physical Parameter Scaling* experiments, we vary the number of unique PDE parameters sampled from the predefined range, varying over 20%, 35%, 50%, 75%, and 100% of the full number of parameters given in Table 1, while using 100% of available initial conditions. Lastly, in our *Temporal Horizon Scaling* experiments (Appendix D.3) we train on solution

frames within 10%, 25%, 50%, 60%, and 75 % of the total time horizon of the simulated trajectories and evaluate on the held-out final 25%. For temporal horizon scaling, we use 100% of the PDE parameter values and initial conditions. Additionally, we investigate how the performance of different architectures behaves as data is scaled along *Initial Condition Scaling* and *Physical Parameter Scaling* (Appendix D.3). We note that *initial condition scaling* and *temporal horizon scaling* experiments are not applicable in the Darcy flow case, as it is a time-independent system (Appendix C.5).

**Model Scaling.** For our model scaling experiments, we logarithmically scale the model size across the total number of parameters: 0.5 million, 5 million, and 50 million parameters. We primarily scale the total number of hidden channel dimensions rather than the model depth to avoid the training difficulties associated with deep neural operators (Tran et al., 2023; Koshizuka et al., 2024; Qin et al., 2024).

# 6 Results

We present selected findings of our evaluation centered on the three fundamental design axes (Section 5). Our baseline model is an FNO with an encoder-downsampler structure (details in Appendix C.1). Complete results can be found in Appendix D.

## 6.1 Optimization

**Data supervision vs. self-supervision.** Figure 2 A–C compares purely data-driven training (Equation 3) with purely self-supervised training using the PDE residual (Equation 4). On all three test sets, direct supervision consistently outperforms self-supervised training, except in cases where both methods produce relative errors close to $\sim 100\%$ (OOD Extreme split of 2D RD). In Appendix Section D.1, we perform additional experiments combining the data loss with the PDE loss, finding that purely data-driven training generally always outperforms any setting with a nonzero coefficient on the PDE loss term. We also perform experiments comparing classical methods such as Newton-CG, L-BFGS-B, and SLSQP against FSNO to better contextualize the results (Appendix Section D.1).

**Test-time training (TTT).** Figure 2 D–E shows the results of performing TTT (optimizing Equation 5) after purely data-driven, supervised training. We consider two settings: 1) training on trajectories resulting from all initial conditions in the dataset, and subsequently evaluating on held-out trajectories from these same initial conditions (Figure 2D), and 2) training on 20% of initial conditions and evaluating on trajectories from unseen initial conditions (Figure 2E). In both figures, we show the improvement in PDE parameter relative loss resulting from TTT across the normalized parameter space (displayed in percentiles). We find that the effectiveness of TTT is generally higher when evaluating on trajectories from unseen ICs and at the extremes of the parameter range, but the degree of improvement is system-dependent. For example, we observe negligible improvements for 2D-NS and 2D-TF, while achieving 10-100% improvements for 2D-RD. The fact that TTT yields greater improvements in OOD initial condition or parameter settings is consistent with our intuition that incorporating information about the mathematical structure of the problem (via the PDE residual) would aid in OOD generalization. In general, since we observe that TTT never appreciably *hurts* performance, we recommend that it be used for PDE inverse problems.

## 6.2 Problem representation

We include model architecture and derivative conditioning results here, deferring temporal conditioning results to Appendix D.2

**Architectural inductive biases.** We compare four architectural inductive biases in a parameter-controlled manner: convolutional (ResNet), spectral (FNO), global attention (scOT) and branch-trunk decomposition (DeepONet). We provide the architectural comparison results in Figure 3A-C and as tables in Appendix D.7. We find that DeepONet performs similarly to ResNet in all settings, see (Figure 3A-C). Since our inverse problems require the model to make a scalar prediction over all co-location points, it is unsurprising that DeepONet and ResNet achieve similar performance, because they both use a ResNet architecture for the branch network. For time-dependent PDEs, we find that FNO generally outperforms ResNet and scOT, particularly in OOD settings (Figure 3A-C). While the architectures differ in factors beyond their inductive biases (e.g., normalization, skip connections, depth), FNO's performance is consistent with the idea that a spectral representation may be well-suited for modeling continuous solution fields. For Darcy Flow, which is time-independent, ResNet and

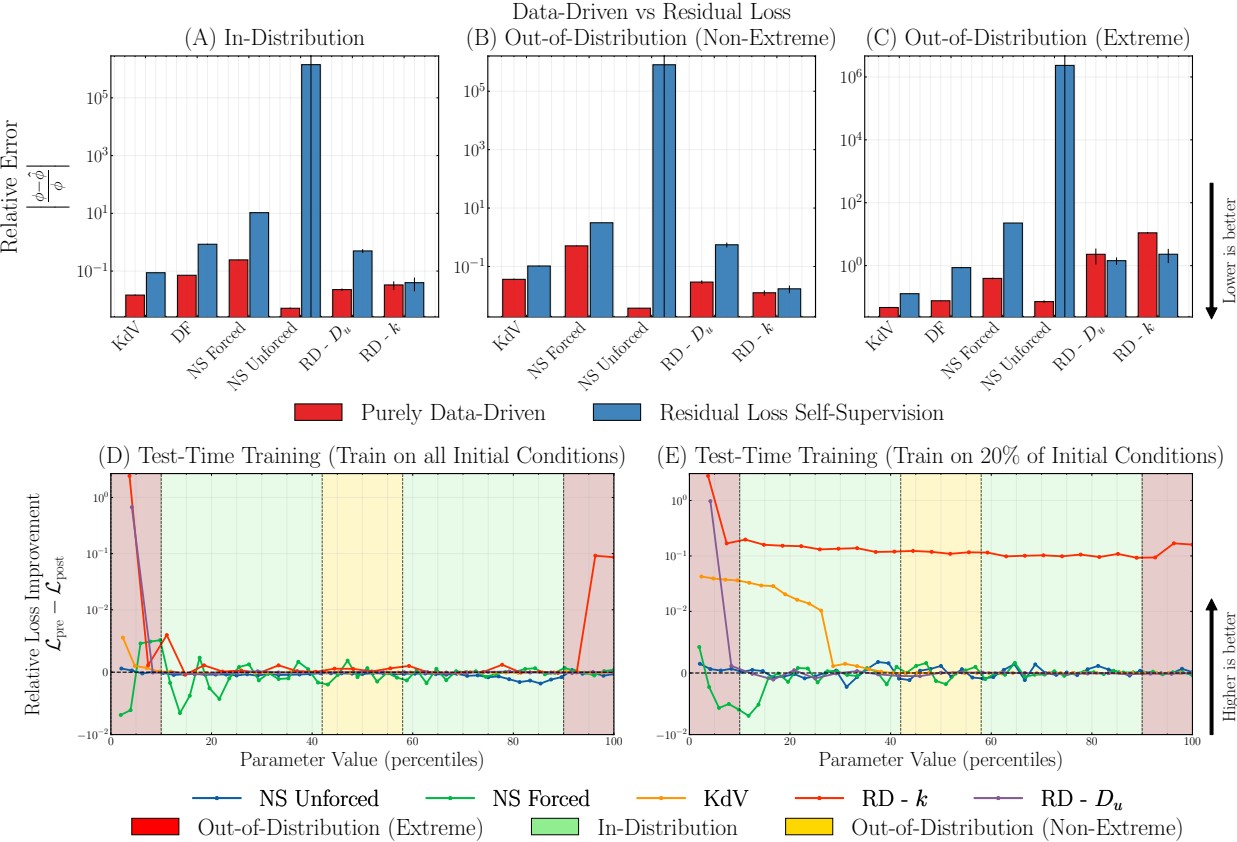

Figure 2: **Optimization approaches for NOs in PDE inverse problems**: (**A**–**C**) The performance of FNO on purely data-driven supervision versus self-supervision using only the PDE residual. For virtually all systems and evaluation settings, purely data-driven supervision consistently outperforms self-supervision using the PDE residual. As expected, performance degrades on the OOD Non-Extreme and OOD Extreme splits. (**D**–**E**) Change in relative error before and after test-time training (TTT) FNO weights, when the initial model is trained on trajectories from all (**D**) or 20% (**E**) of initial conditions in the dataset. Larger values indicate greater improvements from TTT. For the KdV and 2D RD systems, TTT yields noticeable performance improvements in the extreme out-of-distribution parameter regime, and the improvement is more pronounced when performing TTT on unseen initial conditions during test time.

DeepOnet significantly outperform both FNO and scOT (Figure 3A-C). The lack of time-dependency reduces the problem to a spatial modeling task, making it a good fit for convolutional methods.

**Appending partial derivatives.** Figure 3 D–F visualizes the effect of including partial derivatives as additional input features. Across all PDE systems and evaluation settings, models that receive derivative information generally outperform those that receive only the solution fields, demonstrating improved learning efficiency and generalization. While FNO should have the representational capacity to model the spatial derivative operator, we find it helpful to explicitly condition on partial derivatives. In the 2D RD OOD Extreme setting (Figure 3F), both appending and omitting partial derivative information fail to achieve good performance with relative errors $\sim 100\%$.

**Partial observability.** We defer complete partial observability results to Appendix D.5 and D.6, with high-level conclusions stated here. On noisy solution fields, we find that removing partial derivative conditioning improves the robustness of inverse models to degradation, and we find higher sensitivity to Salt-and-Pepper corruption relative to frequency-based corruption. We also find that FNO and ScOT are more robust to non-uniform gridding than ResNet.

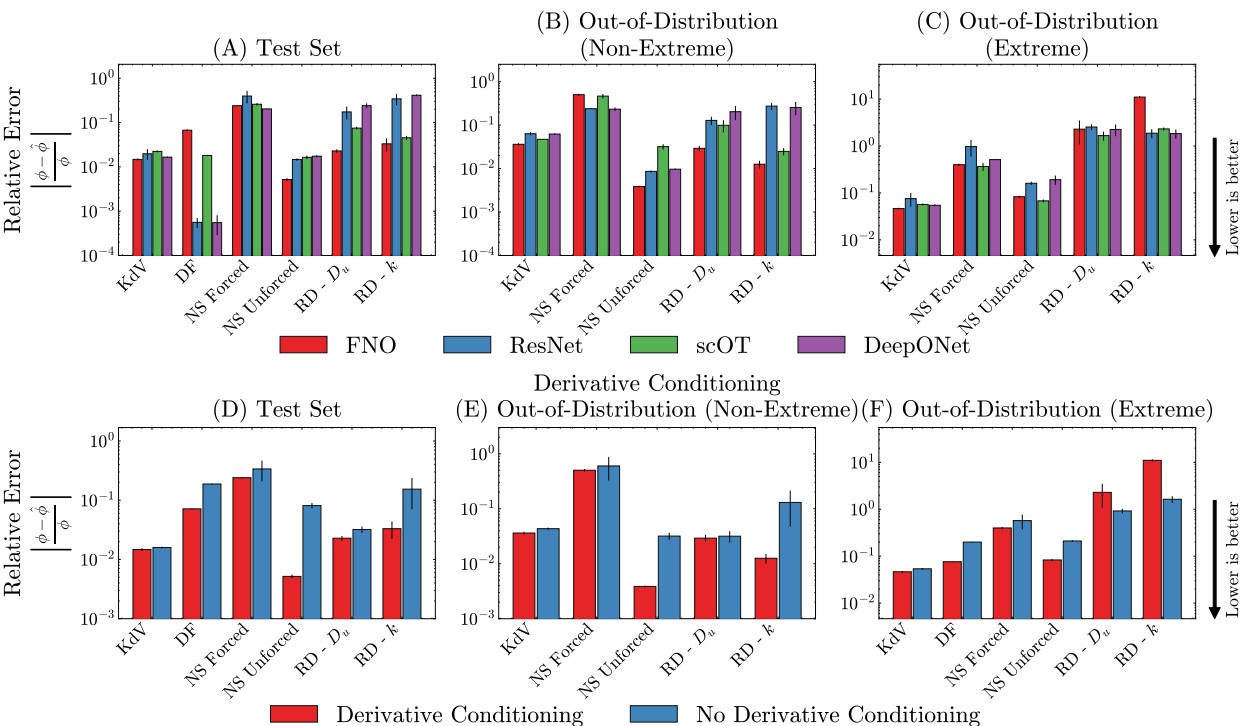

Figure 3: **Impact of problem representation on PDE inverse problem performance.** (**A**–**C**) Comparison of architectural inductive biases (FNO, ResNet, scOT, DeepONet) across evaluation splits. FNO generally outperforms ResNet, scOT and DeepONet on time-dependent PDEs, particularly in In-Distribution and Out-of-Distribution (Non-Extreme) regimes. Results are also shown in tabular form in Tables 6, 7, and 8. (**D**–**F**) Effect of partial derivative conditioning on the FNO architecture, i.e., concatenation versus omission of partial derivatives as features. Conditioning the FNO architecture on partial derivatives consistently outperforms the unconditioned variant.

## 6.3 Scaling

We include data scaling results here, deferring model scaling results to Appendix D.3.

**Initial condition scaling.** We find that scaling the number of initial conditions from 20% to 100% of all initial conditions leads to a clear reduction in error, with every system improving both ID and OOD Non-Extreme errors (Figure 4A,B). The increase in initial conditions yields a modest reduction in relative error for the OOD Extreme case (Figure 4C) and lacks the strong trend seen on other splits.

**Relative effect of data scaling.** To discern which type of data scaling is the most impactful for model performance, we compare the relative effect of each type of data scaling on FNO in Figure 4(D-F). Increasing the number of initial conditions generally yields larger gains in performance compared to increasing the number of PDE parameters. Seeing new physical parameters does not provide the same amount of information as seeing the evolution of new initial conditions with the same physical parameters. In other words, in our setting, new initial conditions provide a better demonstration of the underlying mapping between solution fields and parameters.

## 7 Conclusion

We have introduced *PDEInvBench*, a comprehensive benchmark dataset for PDE inverse problems spanning diverse physical systems. We use *PDEInvBench* to systematically explore and identify best practices for solving PDE inverse problems with neural networks.

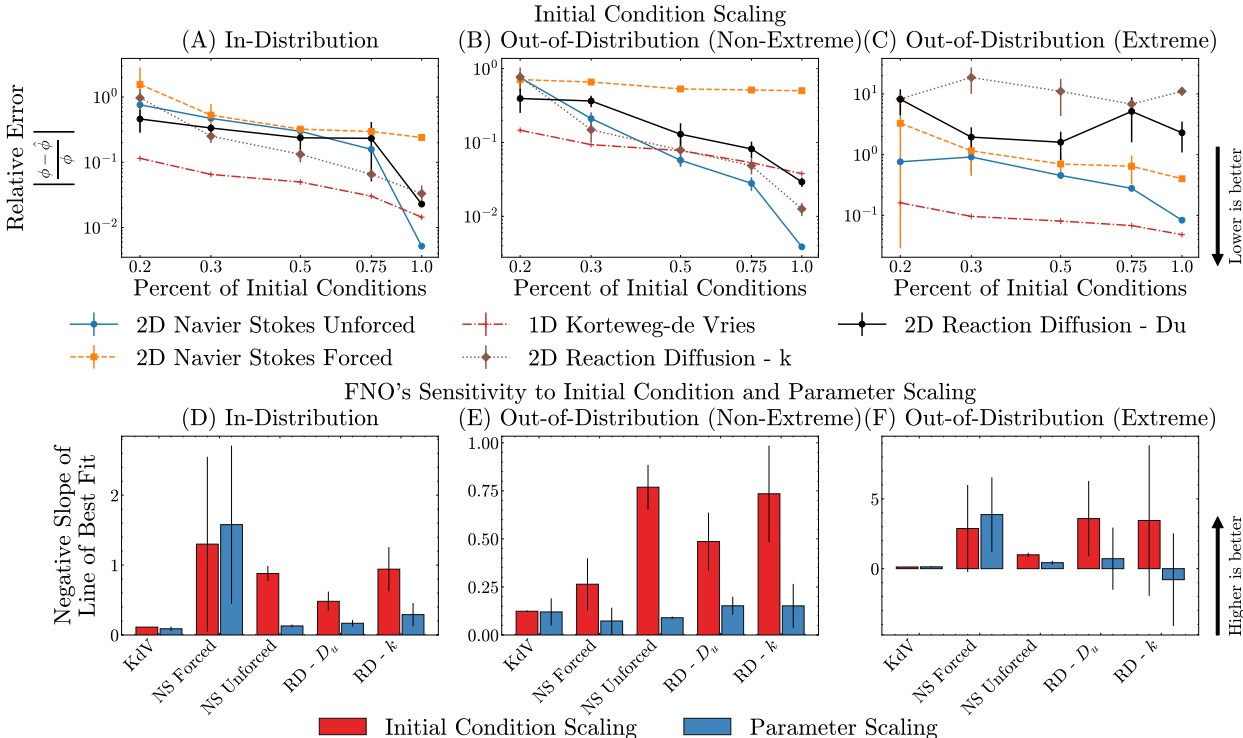

Figure 4: **Dataset scaling with the FNO architecture:** (**A**–**C**) Effect of initial condition scaling on FNO across evaluated systems. Increasing the percent of initial conditions used for training improves performance across all systems and evaluation settings. (**D**–**F**) FNO's sensitivity to initial condition scaling (while using 100% of available PDE parameters) versus scaling the number of generated PDE parameters (while using 100% of the available initial conditions) as measured by the negative slope of the line of best fit (NLS). The NLS is generated by fitting a line of best fit through each line in A–C, alongside parameter scaling, and taking the negative slope. Larger values indicate desirable faster scaling. FNO improves more quickly as the quantity of initial conditions is increased compared to when the quantity of PDE parameters in the training set is increased.

## 7.1 Key Insights

From our investigation of optimization procedures, problem representations, and scaling axes, we distill a set of practical, actionable guidelines for using neural networks to solve inverse problems.

> **Key insight 1:** Practitioners should employ a two-stage training approach when learning neural networks for PDE inverse problems: 1) training with a supervised data loss, and 2) test-time training using the PDE residual to improve generalization to OOD PDE parameters and initial conditions.

Self-supervised training on the PDE residual is consistently worse than purely data-driven supervision (Section 6.1) across systems and irrespective of the weight $\beta$ used for the residual loss term. Additionally, given an initial pre-trained network, test-time training (TTT) using the PDE residual leads to mixed results, with larger improvements when the initial condition and PDE parameter are OOD with respect to the training distribution (Section 6.1). However, since TTT never appreciably hurts performance, we recommend that it be used as long as the application is not highly compute constrained.

> **Key insight 2:** For time-dependent PDEs, the FNO architecture, with its spectral inductive bias, generally outperforms ResNet and Transformer-style architectures. However, given the vast space of architectural factors, more studies should be conducted to better understand how model inductive biases affect predictions for PDE inverse problems.

This, however, does not hold true for time-independent PDEs, like Darcy Flow, where FNO was significantly outperformed by both ResNet and scOT (Section 6.2). Note that our findings are a result of our dataset size and the specific architectures we have chosen. Beyond inductive biases, there are more differences in architectures including normalization, skip connections, depth etc. For the specific systems, dataset regime, and training objective, we found FNO to generally outperform ResNet and scOT.

> **Key insight 3:** Conditioning on the PDE partial derivatives via concatenation consistently improves model performance for the Fourier Neural Operator.

While expressive neural networks should be able to implicitly learn the differential operator, explicitly conditioning on PDE partial derivatives improves performance across all systems (Section 6.2).

> **Key insight 4:** Generating trajectories with more initial conditions per PDE parameter is more beneficial than increasing the number of unique PDE parameters.

This suggests which axis to prioritize when generating data on a strict compute budget. Nonetheless, we find that increasing the amount of data along *any generation axis* universally improves performance (Section 6.3).

## 7.2 Future Work

Future work could extend our benchmarking study in several important directions. One avenue is to consider more PDEs beyond Darcy Flow with coefficients that vary freely in space, testing whether ML models can capture heterogeneous parameter fields rather than just scalar parameters (Leforestier et al., 1991). Another is to investigate inverse problems under sparse, irregularly sampled, or noisy measurements, which would better reflect realistic experimental conditions (Stuart, 2010). While we investigate Butterworth corruption as well as salt and pepper noise in Appendix D.5, noise models are unique to application areas. Future work should explore application-specific noise models and their effects on performance. In these settings, it would be interesting to predict a distribution of PDE parameters with rigorous uncertainty quantification, rather than single point estimates of PDE parameters as done in the present work. Beyond the 2D settings explored here, the inclusion of 3D benchmark datasets such as Navier–Stokes turbulence would further challenge models on higher-dimensional dynamics while also probing scalability under memory and compute constraints (Du and Krishnapriyan, 2025). Our dataset also currently only contains systems with a uniform (square or linear) geometry, while many realistic systems have more complex topologies (e.g., spherical). We plan to include such systems in future work. Additionally, our current benchmark evaluates grid-based architectures; extending the comparison to point-cloud-based methods designed for irregular meshes (e.g., Transolver (Wu et al., 2024), UPT (Alkin et al., 2024)) would require rethinking sections of our evaluation protocol, but is a valuable direction for future work. We simulate what this might look like for partially observed spatial fields in Appendix D.6 but leave a more thorough treatment to future work. Finally, incorporating solver-in-the-loop losses (e.g., running a PDE solver with the predicted parameters and enforcing consistency with a reference trajectory) alongside data and PDE residual losses could enforce stronger physics consistency and improve generalization across regimes and boundary conditions (Um et al., 2020). We believe *PDEInvBench* will act as a solid foundation for such further advances in ML approaches to PDE inverse problems.

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

## A  Neural operator framework

Here we recount the mathematical definition of Neural Operators (NOs) as presented by Li et al. (2021) and Kovachki et al. (2023). NOs are a class of deep learning models which learn mappings between infinite-dimensional function spaces, offering a powerful framework for solving partial differential equations (PDEs). Unlike traditional neural networks, which operate on finite-dimensional Euclidean spaces, neural operators model the non-linear relationship $\mathcal{G}:\mathcal{A}\rightarrow\mathcal{U}$, where $\mathcal{A}$ and $\mathcal{U}$ are Banach spaces of functions defined on bounded sets $D_{\mathcal{A}}\subset \mathbb{R}^{d_{\mathcal{A}}}$ and $D_{\mathcal{U}}\subset\mathbb{R}^{d_{\mathcal{U}}}$, respectively. This is particularly relevant for PDEs, where inputs (e.g., coefficients, initial conditions, and solution fields) and outputs (e.g., PDE parameters, and solution fields) are functions.

The objective is to approximate a nonlinear operator $\mathcal{G}$ using a parameterized NO $\mathcal{G}_\theta$, minimizing the expected error over probability measure $\mu$ on $\mathcal{A}$:

$$\min_\theta \mathbb{E}_{a\sim\mu}[C(\mathcal{G}_\theta(a),\mathcal{G}(a))], \tag{7}$$

where $C:\mathcal{U}\times\mathcal{U}\rightarrow\mathbb{R}$ is a cost function.

Each layer of a NO consists of iterative kernel integration layers, where the $i+1$-th layer update consists of

$$v_{i+1}:=\sigma(\mathcal{W}(v_i)+\mathcal{K}(v_i)), \tag{8}$$

$\sigma$ is a non-linear activation function (e.g., ReLU), $\mathcal{W}$ is a local linear operator, and $\mathcal{K}$ is a nonlocal integral kernel operator. The entire architecture consists of a point-wise lifting operation $\mathcal{P}$, a series of iterative kernel integration layers, and a projection layer $\mathcal{Q}$ (Equation 9).

$$\mathcal{G}_\theta:=\mathcal{Q}\circ\sigma\left(\mathcal{W}^{(L)}+\mathcal{K}^{(L)}\right)\circ...\circ\sigma\left(\mathcal{W}^{(1)}+\mathcal{K}^{(1)}\right)\circ\mathcal{P}. \tag{9}$$

Neural operators are identifiable by two key properties: *discretization invariance*, allowing the same model parameters to be used across different discretizations, and *universal approximation*, enabling them to approximate any continuous operator on compact sets.

## B  Datasets and evaluation metrics

We provide additional details on the datasets that are part of *PDEInvBench*, as well as the evaluation metrics used in our design axis investigation.

### B.1  Datasets.

We study five systems: reaction diffusion, unforced Navier-Stokes, forced Navier-Stokes (turbulent flow), Korteweg-De Vries, and Darcy flow. We also include the break-down of evaluation splits by parameter value in Table 2. For each parameter, a fixed set of initial conditions are evolved according to the dynamics.

**2D Reaction Diffusion (RD).**  We use the following form for the activator $u$ and the inhibitor $v$ coupled system:

$$\partial_t u = D_u\partial_{xx}u+D_u\partial_{yy}u+R_u$$
$$\partial_t v = D_v\partial_{xx}v+D_v\partial_{yy}v+R_v$$

where $R_u$ and $R_v$ are defined by the Fitzhugh-Nagumo equations:

$$R_u(u,v)=u-u^3-k-v$$
$$R_v(u,v)=u-v$$

The parameters of interest are $D_u$ (activator diffusion coefficient), $D_v$ (inhibitor diffusion coefficient), and $k$ (threshold for excitement). We only run experiments on $D_u$ and $k$ due to computational limitations.

**Unforced 2D Navier-Stokes (NS).** We consider the vorticity form of the unforced Navier-Stokes equations:

$$\frac{\partial w(t,x,y)}{\partial t} + u(t,x,y) \cdot \nabla w(t,x,y) = \nu \Delta w(t,x,y), \qquad t \in [0,T], \quad (x,y) \in (0,1)^2 \qquad (10)$$

$$w = \nabla \times u, \quad \nabla \cdot u = 0,$$

$$w(0,x,y) = w_0(x,y), \qquad \text{(Boundary Conditions)}$$

$\nu$ is the physical parameter of interest, representing viscosity.

**Forced 2D Navier-Stokes/Turbulent Flow (TF).** The forced Navier-Stoked equations with the Kolmogorov forcing function are similar to the unforced case with an additional forcing term:

$$\partial_t w + u \cdot \nabla w = \nu \Delta w + f(k,y) - \alpha w$$

$$f(k,y) = -k\cos(ky)$$

Similar to unforced NS, we similarly use the vorticity ($w$) form. $\alpha$, the drag coefficient, and $k=2$, the forced wavenumber, are fixed quantities. $\alpha$ primarily serves to keep the total energy of the system constant, acting as drag. Similar to Kochkov et al. (2021), we use $\alpha = 0.1$. The task is to predict $\nu$.

**Korteweg–De Vries (KdV).** KdV is a 1D PDE representing waves on a shallow-water surface. The governing equation follows the form:

$$0 = \partial_t u + u \cdot \partial_x u + \delta^2 \partial_{xxx} u$$

$\delta$, the physical parameter, represents the strength of the dispersive effect on the system. In shallow water wave theory, $\delta$ is a unit-less quantity roughly indicating the relative depth of the water (Polyanin and Zaitsev, 2012).

**Darcy Flow.** We use the same formulation of Darcy flow as Li et al. (2021). The Darcy flow equations model fluid flow through porous media such as groundwater movement through soil or oil through reservoir rock. The 2-D steady-state Darcy flow equation on the unit box $\Omega = (0,1)^2$ is a second-order linear elliptic PDE with Dirichlet boundary conditions:

$$-\nabla \cdot (a(x)\nabla u(x)) = f(x), \qquad\qquad x \in \Omega, \qquad (11)$$

$$u(x) = 0, \qquad\qquad x \in \partial\Omega, \qquad (12)$$

where $a \in L^\infty((0,1)^2; \mathbb{R}^+)$ is a piecewise constant diffusion coefficient, $u(x)$ is the pressure field, and $f(x) = 1$ is a fixed forcing function.

## B.2 Dataset generation parameters

**Computational requirements.** Some systems are solved using a CPU solver and while others are GPU-accelerated. For the GPU-accelerated systems, we use 40GB A100s, parallelizing over 4 GPUs. We ran the CPU solvers on two AMD EPYC 9354 32-core processors. In the ensuing paragraphs, for each system, we note which type of solver it uses and the approximate wall clock time on a single CPU / GPU machine.

**2D Reaction Diffusion (RD) [parabolic].** Solutions are generated using an explicit Runge-Kutta method of order 5(4) (RK45) for temporal integration, as implemented in the PDEBench framework (Dormand and Prince, 1980; Takamoto et al., 2022), with a relative error tolerance of $10^{-6}$. The spatial discretization employs a Finite Volume Method (FVM) with a uniform grid of $128 \times 128$ cells over the domain $[-1,1] \times [-1,1]$, yielding a cell size of $\Delta x = \Delta y = 0.015625$. The simulations have a burn in period of 1 simulation second. The subsequent dataset simulations span a time interval of $[0,5]$ seconds, discretized into 101 time steps, resulting in a nominal time step of $\Delta t \approx 0.05$ seconds, adaptively adjusted by the RK45 solver to meet the error tolerance. We use SciPy's solve_ivp on CPU, taking $\approx 1$ week to generate.

| PDE | Test | OOD (Non-Extreme) | OOD (Extreme) |
|---|---|---|---|
| 2D RD | $k \in [0.01, 0.04] \cup [0.08, 0.09]$ $D_u \in [0.08, 0.2] \cup [0.4, 0.49]$ $D_v \in [0.08, 0.2] \cup [0.4, 0.49]$ | $k \in [0.04, 0.08]$ $D_u \in [0.2, 0.4]$ $D_u \in [0.2, 0.4]$ | $k \in [0.001, 0.01] \cup [0.09, 0.1]$ $D_u \in [0.02, 0.08] \cup [0.49, 0.5]$ $D_u \in [0.02, 0.08] \cup [0.49, 0.5]$ |
| 2D NS | $\nu \in [10^{-3.8}, 10^{-3.2}] \cup [10^{-2.8}, 10^{-2.2}]$ | $\nu \in [10^{-3.2}, 10^{-2.8}]$ | $\nu \in [10^{-4}, 10^{-3.8}] \cup [10^{-2.2}, 10^{-2}]$ |
| 2D TF | $\nu \in [10^{-4.7}, 10^{-3.8}] \cup [10^{-3.2}, 10^{-2.3}]$ | $\nu \in [10^{-3.8}, 10^{-3.2}]$ | $\nu \in [10^{-5}, 10^{-4.7}] \cup [10^{-2.3}, 10^{-2}]$ |
| KdV | $\delta \in [1.22, 2.48] \cup [3.32, 4.58]$ | $\delta \in [2.48, 3.32]$ | $\delta \in [0.8, 1.22] \cup [4.58, 5]$ |
| 2D DF | Central mass of max-value fraction distribution | - | Tails beyond $\pm 1.5\sigma$ |

Table 2: Parameter splits for different PDEs

**Unforced 2D Navier-Stokes (NS) [parabolic].** We use a pseudo-spectral solver (Du et al., 2024) with a Crank-Nicolson time-stepping scheme (Canuto et al., 2007; Li et al., 2021). The solver is written in Jax (Bradbury et al., 2018) and accelerated using GPUs. Generation takes $\approx 3.5$ GPU days (batch size=32). The solver has a burn in period of 15 simulation seconds, with the next 3 simulation seconds being saved as the dataset. Initial conditions are sampled according to a Gaussian random field (length scale=0.8). The solution is recorded every 1 simulation second and uses a simulation $\partial t = 1e^{-4}$. Solutions are simulated and recorded at a resolution of $256 \times 256$.

**Forced 2D Navier-Stokes (TF).** All solutions in this dataset exhibit turbulence and are generated using a pseudo-spectral solver with a Crank-Nicolson time-stepping scheme (Dresdner et al., 2023). Similar to 2D NS, the solver is written in Jax (specifically leveraging Jax-CFD), and takes $\approx 4$ GPU days (A100). The solver has a burn in period of 40 simulation seconds, with the next 15 simulation seconds being saved as the dataset. The simulator runs at a resolution of $256 \times 256$ and downsamples to $64 \times 64$ before saving at a temporal resolution of $\partial t = 0.25$ simulation seconds.

**Korteweg-De Vries (KdV) [hyperbolic].** The KdV equation is solved on a periodic domain $[0, L]$ using a pseudospectral method with a Fourier basis for spatial discretization ($N_x = 256$ grid points) and an implicit Runge-Kutta method (Radau IIA, order 5) for time integration, implemented via SciPy's `solve_ivp` (Hairer and Wanner, 1999; Virtanen et al., 2020; Brandstetter et al., 2022). Simulations are ran on CPU and takes $\approx 12$ hours. The solver has a burn in period of 40 simulation seconds, with the next 102 simulation seconds being saved as the dataset. The time step is adaptive with absolute and relative tolerances of $10^{-9}$. Like Brandstetter et al. (2022) and Bar-Sinai et al. (2019), initial conditions are sampled from a distribution over a truncated Fourier Series with coefficients $\{A_k, l_k, \phi_k\}_k$.

$$u_0(x) = \sum_{k=1}^{K} A_k \sin\left(2\pi l_k \cdot \frac{x}{L} + \phi_k\right),$$

where $A_k, \phi_k \sim U(0, 1)$ and $l_k \sim U(1, 3)$.

**Darcy Flow (DF) [elliptic].** To complement our generated datasets, we also include Darcy flow from Li et al. (2021). Though we do not generate the Darcy flow data itself, we include relevant parameters as documented by Li et al. (2021) for completeness. The original solver is a second-order finite difference method, with a resolution of $421 \times 421$. Originally written in Matlab, the solver runs on CPU. The lower resolution dataset is generated by downsampling the $421 \times 421$ dataset. The coefficient field $a(x)$ is sampled from $\mu = \Gamma(\mathcal{N}(0, -\Delta + 9I)^{-2})$. $\Gamma$ is a element-wise map $\Gamma: a_i \sim \mathcal{N}(0, -\Delta + 9I)^{-2} \mapsto \{3, 12\}$. $a_i \mapsto 12$ when $a_i \geq 0$ and $a_i \mapsto 3$ when $a_i < 0$. Over the coefficient field, zero Neumann boundary conditions on the Laplacian are enforced.

### B.3 Numerical Convergence

**Grid independence test.** To validate the convergence and correctness of the data we generate, we use the grid-independence test (Lee et al., 2020). The grid-independence test ensures numerical simulation results

are not dependent on the size or resolution of the underlying mesh. To perform the grid-independence test, we compute the Pearson correlation (Schober et al., 2018) and relative $L_2$ loss between solutions generated at the training resolution and a down-sampled high-resolution solution. Similar to the parameter relative loss, the $L_2$ relative loss defined by $RelErr(u_{ref}, u) = \frac{||u - u_{ref}||_2}{||u_{ref}||_2}$. To downsample the high resolution solutions, we use nearest neighbor interpolation. We report results on both OOD-Extreme splits separately, represented by "small" (lower parameter values) and "large" (higher parameter values) in table 3. The OOD-Extreme splits capture the smallest and largest parameter values, probing whether solutions are converged at extreme ends and provide a principled way to test for convergence. High resolution solutions for 2D reaction diffusion are generated at $512^2$ and at $256^2$ for unforced 2D Navier-Stokes. Both 2D reaction diffusion and unforced 2D Navier-Stokes exhibit low relative error and high correlation in both parameter regimes, indicating convergence. The chaotic nature of turbulence means the grid independence test is not a good measure of convergence for forced 2D Navier-Stokes, and we instead defer to the energy spectrum analysis in B.4.

| System | $L_2$ Rel. Err. (%) | Pearson Corr. | $L_2$ Rel. Err. (%) | Pearson Corr. |
|---|---|---|---|---|
| *Split* | *OOD-Extreme Small* | | *OOD-Extreme Large* | |
| 2D RD - $k$ | $3.2611\% \pm 0.8585\%$ | $0.999874 \pm 1.26e^{-4}$ | $2.1440\% \pm 0.1016\%$ | $0.999857 \pm 1.42e^{-4}$ |
| 2D RD - $D_u$ | $2.6687\% \pm 0.8578\%$ | $0.999836 \pm 1.18e^{-4}$ | $2.7365\% \pm 0.7957\%$ | $0.999896 \pm 1.43e^{-4}$ |
| Unforced 2D NS | $2.1656\% \pm 0.0002\%$ | $1.008181 \pm 2.45e^{-4}$ | $2.1303\% \pm 0.0000\%$ | $1.008066 \pm 3.e^{-6}$ |

Table 3: Grid independence test for 2D reaction diffusion and unforced 2D Navier-Stokes partitioned by parameter regime.

### B.4 Numerical Convergence of Forced 2D Navier-Stokes

While Dresdner et al. (2023) discuss and verify the convergence of forced 2D Navier-Stokes, they do not cover the broad range of parameters we generate data for. Our dataset includes Reynolds numbers outside the range of parameters they study. We perform a similar analysis by comparing the energy spectra relative to a high resolution reference solution. The reference solution is generated with a $2048^2$ spatial grid over 1 simulation second. While we generate training and evaluation data for a longer length of time, the chaotic nature of the problem means errors will diverge (Budanur and Kantz, 2022) for sufficiently long simulations.

**Energy spectrum.** An important method for understanding the convergence of numerical simulations, particularly pseudo-spectral methods for turbulence, is to examine the resulting energy spectra (Canuto et al., 2007; Evans, 2022). The energy spectra is an important tool for understanding the dynamics of turbulence and describes the energy distribution across different spatial length scales. In Figure 5 we plot the energy spectra of two viscosities, representing simpler turbulence ($\nu = 5e^{-3}$) and the most turbulence captured by our parameter range ($\nu = 1e^{-5}$) For both parameter values, we see convergence into the dissipative range characterized by a steep drop-off in energy followed by mild oscillations, highlighted via the blue bounding box.

At the most turbulence regimes (e.g., $\nu = 1e^{-5}$), generating and storing solutions at $2048^2$ is necessary to capture all fine features. We include a small subset of high resolution data as a separate evaluation for downstream users of *PDEInvBench*. However, we keep our main text evaluations on a $64^2$ grid as it is a harder problem than predicting on a finely discretized grid. With a sufficiently fine discretization that resolves the dissipative cutoff, the viscosity is strongly constrained by the smallest active length scales. $\nu$ directly controls the high-wavenumber energy drop-off and the onset of the dissipative range in the energy spectrum. In contrast, evaluations on a coarse grid should be viewed as a partial observation of the flow after a low-pass filter or downsampling projection. For the inverse problem, we infer $\nu$ from the observation where the downsampling removes the small-scale content that is most informative about viscous dissipation. When the dissipative range is only partially observed, different viscosities can induce very similar large-scale trajectories and spectra over the resolved band, making parameter identification ill-posed and therefore a strictly harder than when the full field is available. At $64^2$, our evaluation should be interpreted as a partially observed inverse problem. The model is given only the large-scale components of a resolved simulation, and is required to infer parameters from incomplete scale information, providing a test of robustness.

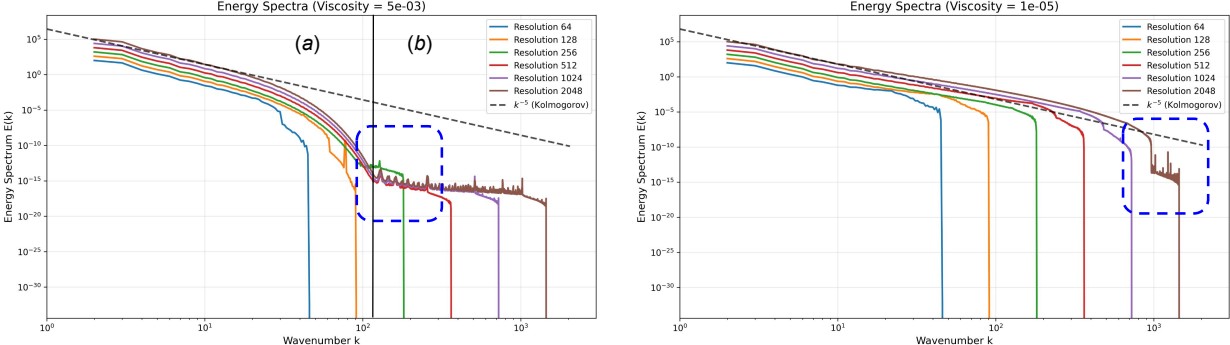

Figure 5: **Energy spectra convergence of 2D Navier-Stokes Forced.** (Left) Energy spectra for $\nu = 5e^{-3}$. Region $a$: inertial range, with rough adherence to the $k^{-5}$ Kolmogorov power law. Region $b$: dissipative region where energy leaves the system. (Right) Energy spectra for $\nu = 1e^{-5}$. (Both) The blue bounding box highlights convergence behavior, characterized by a steep drop-off in energy spectra into minor oscillations. For $\nu = 5e^{-3}$ multiple resolutions exhibit convergence including $256^2$. In the high turbulence regime ($\nu = 1e^{-5}$), $2048^2$ captures the full energy spectra.

### B.5    Additional information on evaluation splits

We take the best-performing models from our validation set and evaluate on the ID Test, OOD (Extreme) and OOD (Inclusive) at test time. This partitioning strategy allowed us to assess model performance both within the training distribution and at different levels of extrapolation. In the case of 2D reaction diffusion with multiple physical parameters, we generalize the notion of parameter ranges on a line to a hyper-cube (Figure 6).

For Darcy flow, we compute the fraction of grid points in each coefficient field that take the maximum value (12). This statistic is approximately normally distributed across the dataset. We partition coefficient fields according to this distribution using $\pm 1.5$ standard deviations, with the central mass defining the in-distribution (ID) test set and the tails defining the OOD-Extreme setting.

We also include the specific parameter ranges of each split for each dataset in Table 2.

### B.6    Evaluation metrics

**Error bars for relative error**    For each of the three seeded runs, we select the model with the best performance on the validation set. We compute the relative error for each model on the evaluation splits. The mean and standard deviation of these three relative errors are calculated and reported as the error bars in our plots (1 $\sigma$ error bars).

**Error bars for negative slope of the line of best fit**    Using the best-performing models from the validation sets of the three seeded runs, we compute the negative slope of the line of best fit (NLS) for each seed and data scaling procedure. For each scaling procedure, we calculate the mean and standard deviation of the three NLS values and report these as the error bars in our plots (1 $\sigma$ error bars).

For our evaluation splits, we deterministically sample every 10th window of PDE frames along the generated trajectory.

## C    Methodology for Investigating Design Axes

We now provide additional details on our methodology for investigating the three key design axes considered in our work: 1) optimization procedure, 2) problem representation and inductive bias, and 3) scaling properties. In this section, we focus on the methodology for experiments included in the main text. We include methodology and results for additional experiments in Section D.

### C.1    General details

We first provide details shared across all design axes.

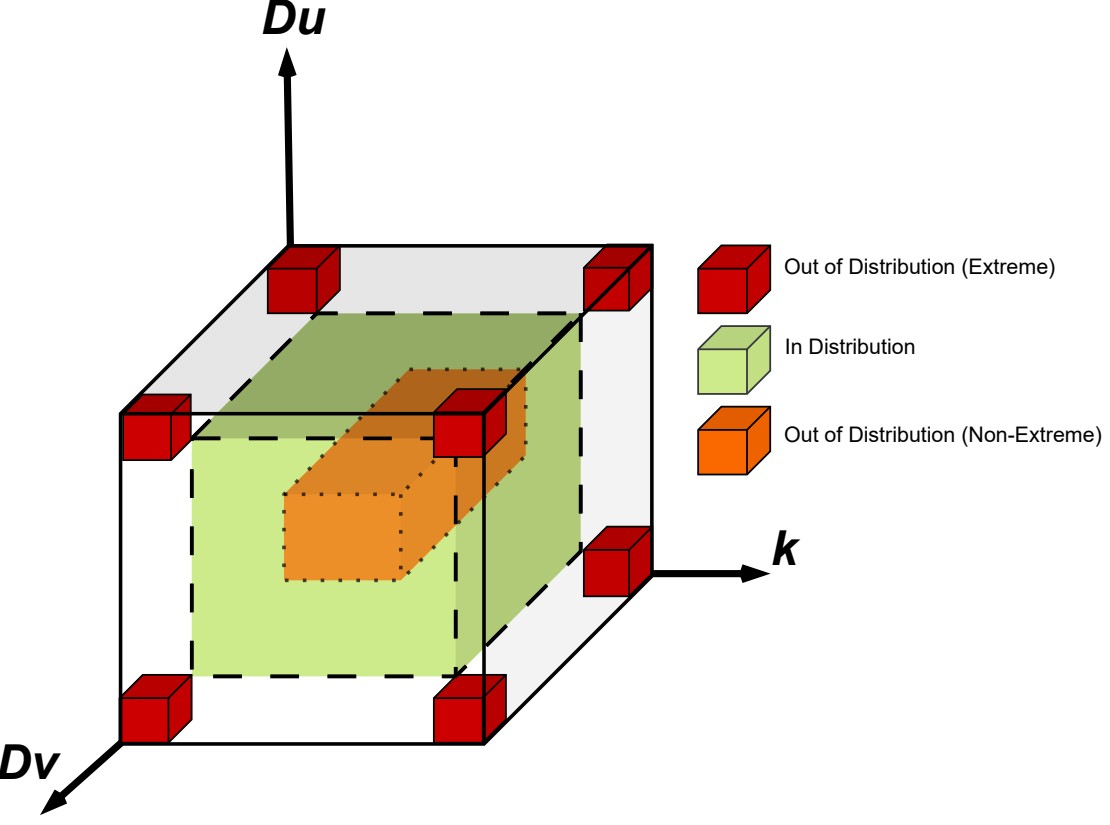

Figure 6: **Parameter partitioning for 2D RD.** The parameter space for 2D RD forms a cube with each dimension corresponding to possible values along a parameter. The inner cube (orange) with each edge covering the middle 16% of parameter values corresponds to the OOD (Non-Extreme) split. The middle cube (green) with each edge covering 32 % of parameters corresponds to the ID setting. The corner cubes (red) with edge length covering 10% of parameter values correspond to the OOD (Extreme) setting.

### C.1.1 Base model architecture

The base model for solving the inverse problem is a neural network designed to predict PDE parameters from input data, consisting of three primary components: a FNO-based encoder, a convolutional downsampler to reduce the dimensionality of the activation maps, and a MLP regression head to predict the PDE parameter. The encoder processes 2 temporal PDE frames of input data, incorporating partial derivatives specific to the PDE. The encoder consists of four FNO layers, 16 Fourier modes per spatial dimension, and hidden channel dimension of 64. The encoder output is passed to a four-layer convolutional downsampler. Each downsampler layer applies a two-dimensional convolution with 64 input and output channels, a kernel size of 3, a stride of 1, and padding of 2, followed by ReLU activation and two-dimensional max-pooling with a kernel size of 2. Thus, the downsampler reduces the spatial resolution by a factor of 16. The downsampler output is flattened and fed into an MLP head with one hidden layer of 64 units using ReLU activation. A single value is returned, corresponding to the PDE parameter.

**DeepONet.** Our DeepONet implementation follows the standard branch-truck decomposition with slight changes adapting it for the inverse problem setting. The branch network, which processes the PDE solution at collocation points, takes in the full spatial solution and is implemented as a ResNet, primarily for computational

efficiency when handling the full grid. The trunk network for encoding sensor locations is implemented as a pointwise MLP with residual connections and GeLU activations. Since most of our inverse problems predict a scalar PDE parameter rather than a spatio-temporal field, we apply mean pooling over the pointwise branch-trunk inner produce to produce the final scalar output.

### C.1.2 Training details

We provide training details and hyperparameters common to all systems and experiments. We use a batch size of 32. We use the Adam optimizer with a cosine decay learning rate scheduler. The initial learning rate is set to 1e-4, and is decayed to 0. We train our models for 200 epochs.

### C.1.3 Compute usage

Here we provide the total training time, and memory usage of all datasets and model performed on NVIDIA RTX A100 in Table C.1.3.

| Dataset | Model | Total Training Time | Per Batch Inference Speed | GPU Memory Usage |
|---|---|---|---|---|
| | FNO | 1 hour | 33.64 it/s | 4.7 GB |
| 2DNS | SCOT | 2.5 hours | 11.15 it/s | 4.2 GB |
| | ResNet | 1.5 hours | 22.07 it/s | 3.8 GB |
| | FNO | 6.5 hours | 1.95 it/s | 8.5 GB |
| 2DRD | SCOT | 6.45 hours | 1.73 it/s | 4.5 GB |
| | ResNet | 6.4 hours | 1.71 it/s | 8 GB |
| | FNO | 1 hours | 20.14 it/s | 3.3 GB |
| 2DTF | SCOT | 1.2 hours | 12.6 it/s | 2.3 GB |
| | ResNet | 1.7 hours | 7.86 it/s | 6.3 GB |
| | FNO | 33 minutes | 32.35 it/s | 4.7 GB |
| 1DKDV | SCOT | 1.5 hours | 23.6 it/s | 5.8 GB |
| | ResNet | 46.68 minutes | 23.6 it/s | 4.2 GB |
| | FNO | 18 minutes | 9.5 it/s | 7.5 GB |
| 2DDF | SCOT | 29 minutes | 3.45 it/s | 4.5 GB |
| | ResNet | 60 minutes | 1.1 it/s | 8 GB |

Table 4: Training time, inference speed, and memory usage of FNO, SCOT, and ResNet across different datasets.

## C.2 Optimization procedure

We provide additional details on the optimization procedure design axis.

### C.2.1 Test-Time Training details

For test-time training (TTT), we consider two configurations: *per-batch* tailoring, in which we adapt the model for each batch, and *per-element* tailoring, where we finetune individually for each test sample. In both cases, we perform tailoring for 50 gradient steps with a learning rate of 1e-5 and the adam optimizer. We show the results of *per-element* tailoring in 6. Here, we include comparisons of *per-batch* tailoring and *per-element* tailoring. We also include ablations on varying the anchor loss weight $\alpha \in \{0, 0.01, 1\}$.

## C.3 Problem representation and inductive biases

We provide additional details about the experiments relating to the problem representation and inductive biases design axis.

### C.3.1 Model hyperparameters and details

We provide hyperparameters for all model variants considered. For all model variants, only the encoder portion of the model is changed. The convolutional downsampler and MLP head remain the same as what was described in Section C.1.1.

**Fourier Neural Operator (FNO)**   The FNO architecture is configured with 4 Fourier layers, 16 modes, and 64 hidden channels.

**Scalable operator Transformer (scOT)**   The scOT architecture employs 4 Swin Transformer layers with an embedding dimension of 36, patch size of 4, varying attention heads (3, 6, 12, 24), and a hidden size of 32.

**ResNet** The ResNet architecture features 6 residual layers with 128 hidden channels.

When predicting a single scalar parameter, each model is followed by a convolutional downsampler and single-layer MLP. When predicting a spatial parameter field as in Darcy-Flow, the convolution downsampler is replaced with an identity map. The MLP head is replaced with a 3 layer convolution head that uses point-wise convolutions to generate a segmentation map. The final activations are converted to a binary segmentation map using the sigmoid function.

### C.4 Data scaling

We provide additional results on the data scaling experiments, deferring a description of model scaling to Section D.3.1.

We investigate how the amount of training data affects performance by systematically varying three aspects of the dataset:

1. **Initial condition scaling**: We vary the number of initial conditions per parameter value at {20%, 35%, 50%, 75%, and 100%} of the full dataset.

2. **Parameter scaling**: We vary the density of parameter sampling at {20%, 35%, 50%, 75%, and 100%} of the full range.

3. **Temporal scaling**: We vary the total temporal horizon on which the model is trained by varying the sampled frames from the first {10%, 20%, 50%, 75%} of the total generated temporal range of our dataset. The evaluation set is the final 25% of the generated temporal range for the in-distribution test setting and the entire temporal trajectory for the OOD evaluations settings.

We also investigate how scaling the amount of training data along initial conditions and parameters impacts performance across the different architectures. When the percentage-based data scaling results in non-integer number of total frames, we round up to the nearest integer.

### C.5 Darcy Flow experimental constraints

Since Darcy flow is time-independent, initial-condition scaling, temporal-horizon scaling, and temporal conditioning experiments are not applicable to the system.

## D Additional results

We provide the results of additional experiments not included in the main text. Again, we split these into the same three principle design axes.

### D.1 Optimization

### D.1.1 Benchmarking classical methods against Neural Operators

We compare FNO to the following classical baselines Newton-CG, L-BFGS-B, SLSQP. For solvers requiring an initial estimate, we uniformly sample 10 initial parameters (or log-spaced uniformly for log-spaced parameters) from within the parameter range. Across all systems, the classical methods we try perform worse than the FNO baseline, suggesting the viability of Neural Operators for PDE inverse problems Table 5.

| System | L-BFGS-B | Newton-CG | SLSQP | FNO |
|---|---|---|---|---|
| 2D RD $- k$ | $1.6 \pm 6.0 \times 10^{-4}$ | $1.6 \pm 3.7 \times 10^{-9}$ | $1.6 \pm 2.0 \times 10^{-4}$ | $\underline{0.35 \pm 0.021}$ |
| 2D RD $- D_u$ | $0.98 \pm 9.8 \times 10^{-9}$ | $0.98 \pm 2.1 \times 10^{-15}$ | $0.98 \pm 2.0 \times 10^{-10}$ | $\underline{0.15 \pm 0.067}$ |
| 2D NS Unforced | $5.1 \pm 7.2$ | $5.1 \pm 7.2$ | $5.2 \pm 7.1$ | $\underline{0.021 \pm 0.0011}$ |
| 2D NS Forced | $31 \pm 50$ | $9.1 \pm 7.7 \times 10^{-13}$ | $32 \pm 49$ | $\underline{0.37 \pm 0.011}$ |
| 1D KdV | $0.092 \pm 0.0049$ | $0.090 \pm 0.0012$ | $0.21 \pm 0.025$ | $\underline{0.03 \pm 0.0016}$ |

Table 5: Comparison of classical optimization methods and FNO across PDE systems (mean ± std). Comparing FNO with L-BFGS-B, Newton-CG, and SLSQP. The classical optimization methods are initialized with 10 uniformly sampled seeds from the parameter range. FNO consistently achieves the lowest error across all systems.

### D.1.2 Physics-Informed Loss Function

In addition to the loss formulations considered in the main text, we consider a generalized, "physics-informed" loss combining data-driven supervision with self-supervision from the PDE residual:

$$\mathcal{L}_{\text{physics-informed}} = \alpha \mathcal{L}_{\text{data}} + \beta \mathcal{L}_{\text{res}}. \tag{13}$$

Here, $\alpha, \beta$ are weighting coefficients. Note that this is a physics-informed loss for the inverse problem, meaning $\mathcal{L}_{\text{data}}$ is taken with respect to the ground truth PDE parameter $\phi$ and predicted parameter $\hat{\phi}$ (e.g., $\mathcal{L}_{\text{data}} = ||\hat{\phi}^i - \phi^i||_2 / ||\phi^i||_2$). Section 6.1 presented results on two extremes: a supervised, purely data-driven setting ($\alpha = 1, \beta = 0$), and a purely self-supervised, residual-driven setting ($\alpha = 0, \beta = 1$). Building on this, we vary the residual weight $\beta$ used in the physics-informed loss (Equation 13) from 0 to 1, showing results in Figure 7.

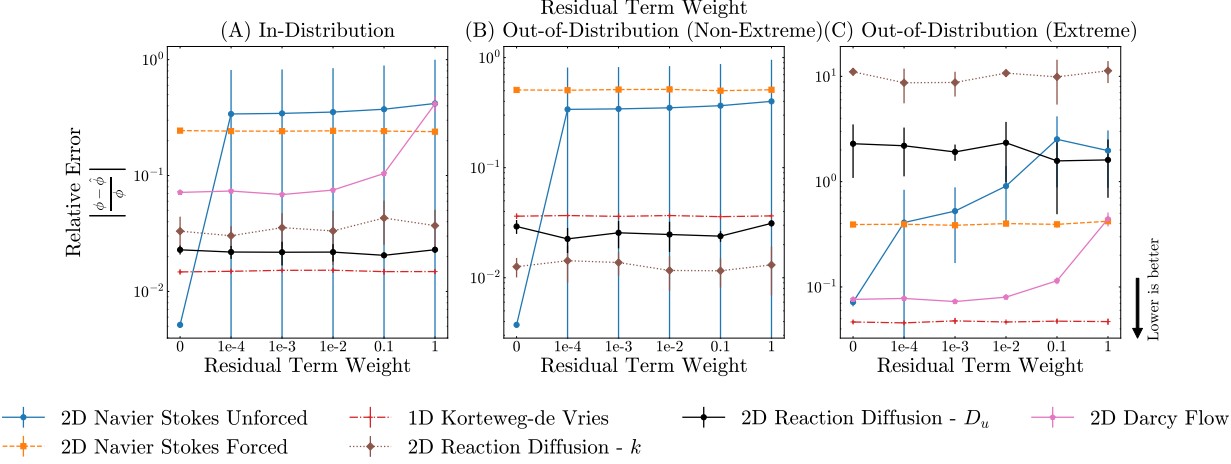

Figure 7: **Effect of PDE residual loss weights in the PINNs loss.** Relative error from the best performing models on the validation set across three scenarios: ID (left), OOD Non-Extreme (middle), and OOD Extreme (right). The x-axis represents PDE residual term weight ranging from 0 to 1 on a logarithmic scale. Joint training with the PDE residual offers no significant improvements over direct parameter supervision, as all nonzero weights yield worse performance than with a weight of zero.

Across all evaluation settings and PDE systems, we observe that no nonzero value of the residual weight significantly outperforms basic parameter supervision alone ($\beta = 0$). Most systems show relatively flat performance curves as $\beta$ increases from 0 to 0.1, followed by performance degradation with higher weights. The residual term becomes actively detrimental at higher weights, likely because it creates a more complex loss landscape that complicates optimization Krishnapriyan et al. (2021). These findings suggest that if paired data is available, practitioners may safely omit the residual term during initial training without sacrificing performance.

### D.1.3 Varying anchor weight in test-time training

Figure 8 demonstrates how varying the anchor loss weight in the test-time training objective ($\alpha$ in Equation 5) influences performance. We find that the optimal setting uses equal weighting between residual and anchor terms. When the anchor loss is weighted too lightly, the relative error tends to increase with training steps, indicating optimization instability.

### D.1.4 Per-element vs per-batch tailoring

We compare performing TTT on a *per-element* basis (batch size of 1) versus a *per-batch* basis (batch size of 32) with anchor loss weights of 1 and show results in Figure 9.

TTT *per-batch* generally perform the same *per-sample* in all evaluation settings.

### D.1.5 Test Time Tailoring Comparison with varying levels of ICs

We compare the performance of test-time training on models trained on 20% of total available initial conditions and 100% of initial conditions by system in Figure 10 (this is a per-system view of Figure 2). We see that

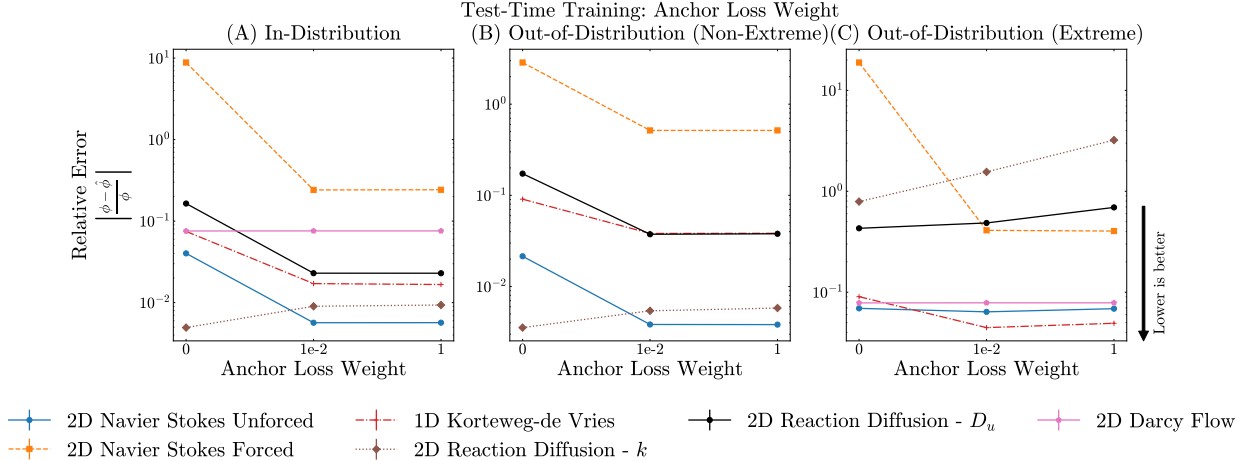

Figure 8: Effect of test time tailoring with different anchor loss weights per batch. Using an anchor loss weight equal to the PDE residual loss weight (1) generally gives the best performance

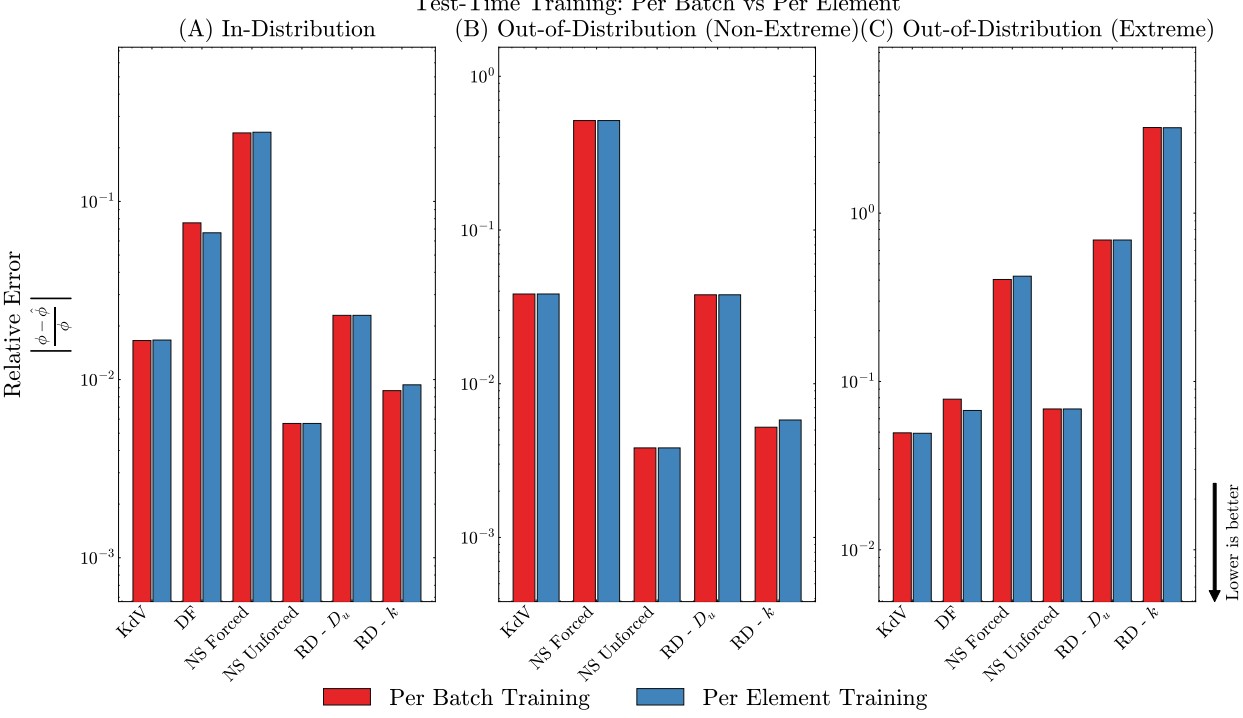

Figure 9: Comparing *per-batch* vs *per-batch* test time training. Both settings generally perform the same across systems and evaluation settings.

test-time-tailoring generally helps in the case of limited number of iniital conditions and out of distribution parameter regimes parameter regimes. It never appreciably reduces performance. Therefore, we recommend performing TTT for inverse parameter estimation, especially over batches of input frames.

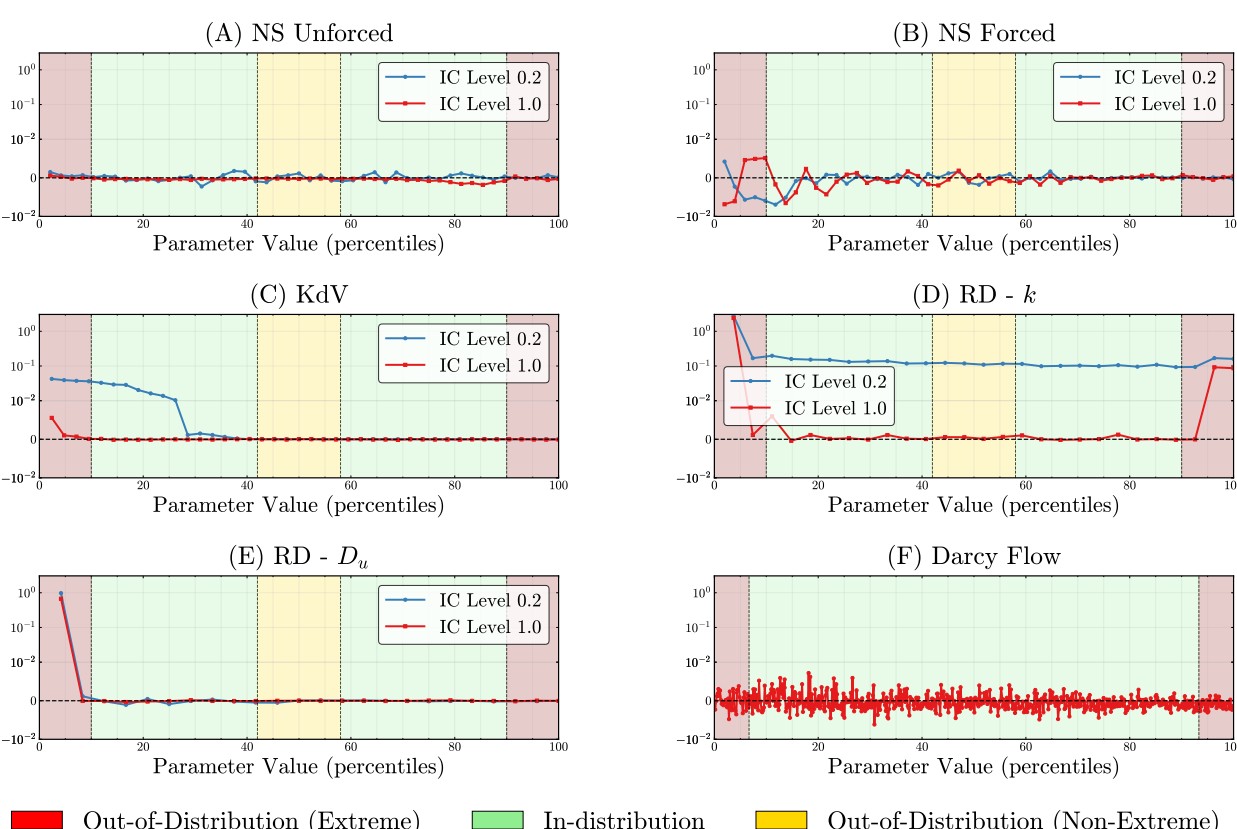

Figure 10: Evaluating test-time training with varying quantities of initial conditions during training time and across parameter regimes. For the KdV and 2D RD systems, TTT yields noticeable performance improvements in the extreme out-of-distribution parameter regime, and the improvement is more pronounced when performing TTT on unseen initial conditions during test time.

## D.2 Problem representation

### D.2.1 Varying number of temporal conditioning frames

We vary the number of conditioning frames from the ground truth solution trajectories that are passed as input to the inverse model. We vary the number of input conditioning frames across {2,5,10,15,20} for each system. For each configuration, we train separate models while maintaining consistent architecture and optimization settings. This experiment is not applicable to Darcy Flow since it is a time-independent PDE (Appendix C.5). Results are shown in Figure 11.

Unlike appending partial derivative information, which shows consistent benefits as demonstrated in Section 6.2, the optimal number of conditioning frames varies significantly across PDE systems. The Navier-Stokes unforced system generally shows better performance with more frames (10-20), while KdV attains optimal performance around 10 frames. The 2D reaction-diffusion parameters (k, Du, Dv) show less sensitivity to frame count in most evaluation settings. This system-specific variability highlights the importance of empirical tuning of this hyperparameter for each PDE family rather than assuming a universal "more is better" approach.

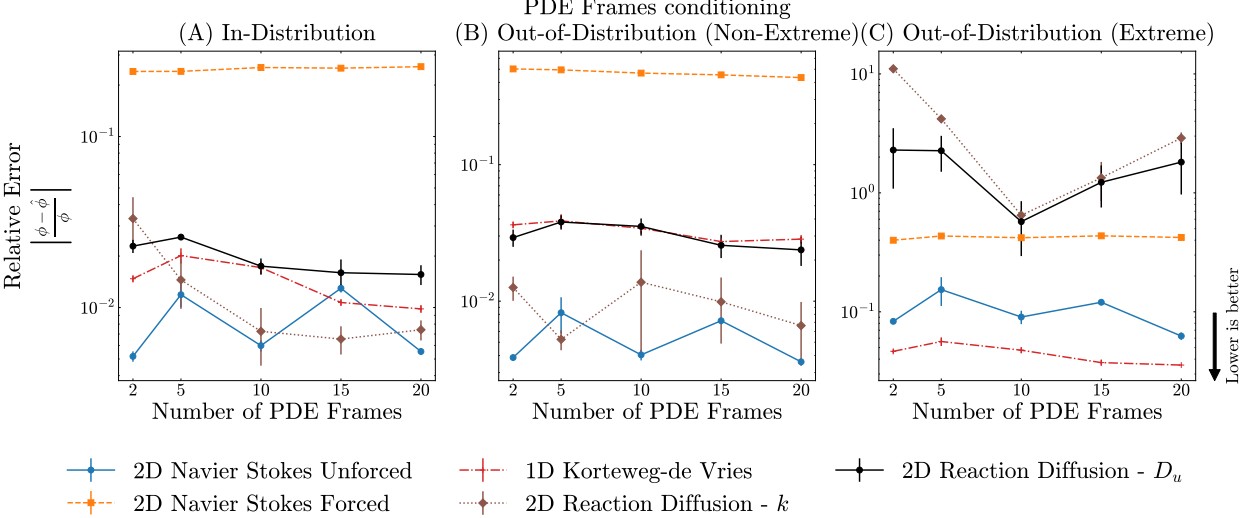

Figure 11: **Varying temporal conditioning frames.** Comparison of inverse problem performance with increasing number of temporal frames across evaluation settings for all time dependent settings. There is no consistent trend in scaling the conditioning PDE frames and inverse problem performance for FNO.

The lack of consistent improvement with more frames may reflect redundancy in temporal information beyond a certain point or increased input dimensionality complicating the learning process.

### D.3 Scaling

### D.3.1 Model scaling

To investigate the scaling behavior of neural operators for inverse PDE problems, we conducted a systematic model scaling experiment. We scaled the model size logarithmically across three configurations: 0.5M, 5M, and 50M parameters, focusing primarily on increasing width rather than depth. Our scaling strategy deliberately avoided increasing model depth beyond 4-6 layers due to the known training difficulties associated with deeper neural operators. Instead, we significantly increased the hidden channel dimensions from 16 to 64 to 200 as we moved from 0.5M to 5M to 50M parameters. All models share the same core architecture: an FNO encoder with 16 modes, a convolutional downsampler with identical kernel configurations (3×3 kernels, stride 1, padding 2), and a consistent MLP design with ReLU activation. Importantly, we maintained a relatively constant ratio between the parameter count of the main FNO encoder and the downsampler networks across all scale configurations. We present results in Figure 12.

As a general trend, we observe that ID and OOD non-extreme performance generally improves with increase model size (Figure 12A and B). However, this trend does not hold with the OOD non-extreme split. This suggests that bigger models show better interpolative performance but do not guarantee better extrapolation.

### D.3.2 Data scaling

In addition to scaling the number of initial conditions in the dataset, we also scale the number of unique PDE parameters and the length of the ground truth trajectories to which the model has access.

**Effect of scaling initial conditions across architectures** More initial conditions generally lead to better performance for all architectures in most evaluation settings and systems, see Figure 13. FNO generally exhibits greater data efficiency when scaling up the number of initial conditions compared to ResNet and scOT.

**Scaling the number of PDE parameters.** We investigate the effect of reducing the fraction of unique PDE parameters included in the training dataset. We remove the selected fraction of parameters from the training set and uniformly sample frames from the remaining parameters.

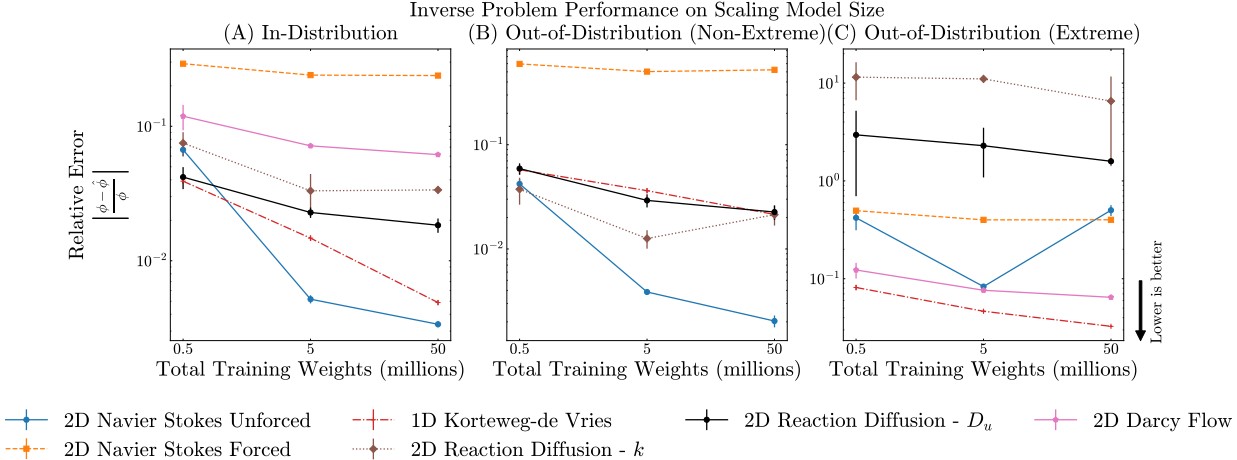

Figure 12: Inverse Problem Performance as a function of model size. Performance curves for different PDE systems across test scenarios as the number of trainable parameters increases from 0.5 million to 50 million. Model performance generally improves in all evaluation regimes when increasing the total number of model parameters.

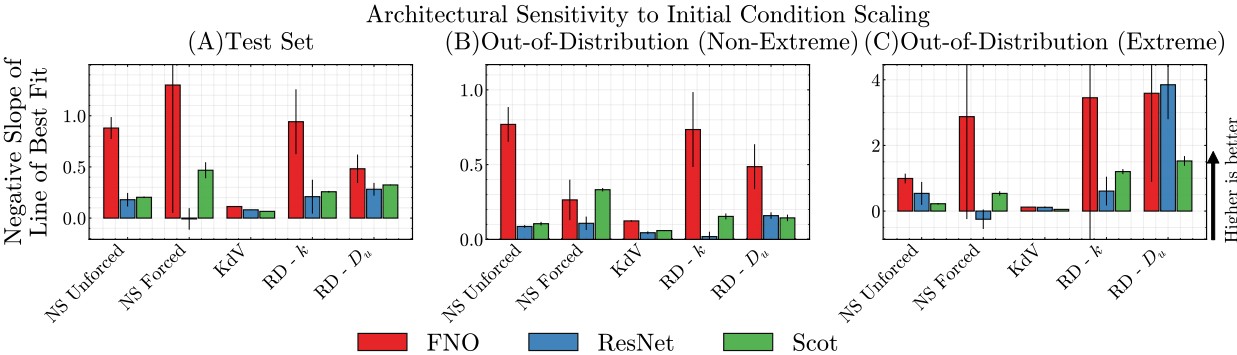

Figure 13: **Effect of initial condition scaling across architectures** Evaluating data efficiency of different architectures to scaling the number of initial conditions. Increasing the total number of initial conditions during training improves performance for all architectures, with FNO being the most data efficient.

Increasing the number of parameters generally leads to better performance in all evaluation settings for most systems. However, as demonstrated in Figure 4, the improvements are not as large as those resulting from increasing the number of initial conditions per PDE parameter.

**Effect of scaling PDE parameters across architectures** More PDE parameters generally lead to better performance for all architectures in most evaluation settings and systems Figure 15.

**Scaling the length of generated trajectories** As another way to investigate data scaling properties, we vary the total time horizon of the ground truth trajectories used for training, leaving out the final 25% of the in-distribution trajectories. We show results in Figure 16. This experiment is not applicable to Darcy Flow since it is a time-independent PDE (Appendix C.5).

Longer training time horizons generally lead to better performance for neural operators in the in-distribution setting, as shown in Figure 16(A) where relative error decreases as the percentage of PDE parameters increases. However, this general pattern does not hold in any of the out-of-distribution settings Figure 16 (B, C). Therefore, training on longer temporal trajectories does not lead to representations that perform better on unseen parameters in either non-extreme or extreme out-of-distribution test cases. This suggests that extending

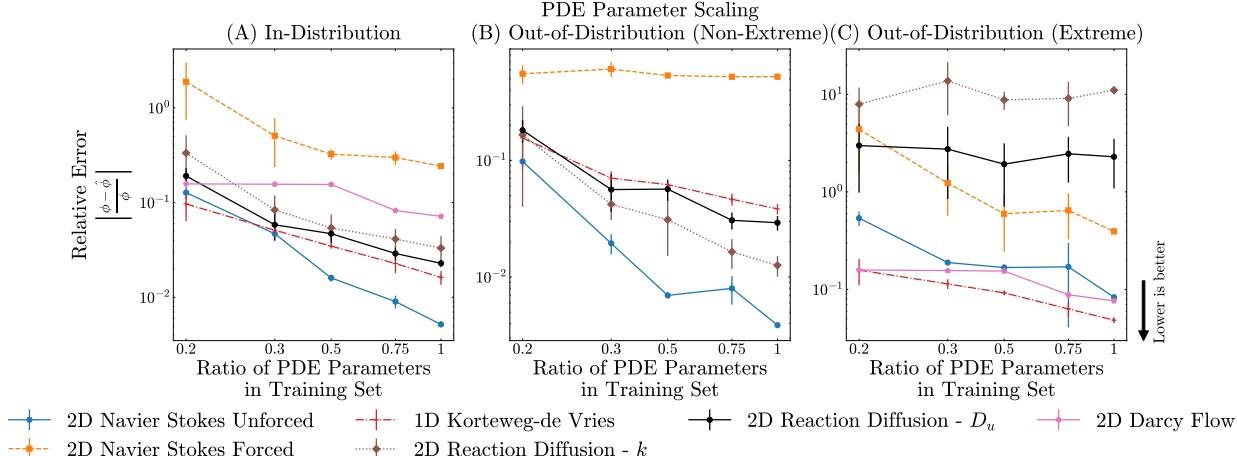

Figure 14: **Effect of scaling the total number of generated PDE parameters.** Evaluating inverse problem performance on different quantities of available data by scaling the total number of generated parameter settings of the PDEs. Increasing the number of training trajectories along generated PDE parameters improves test time performance on unseen parameters.

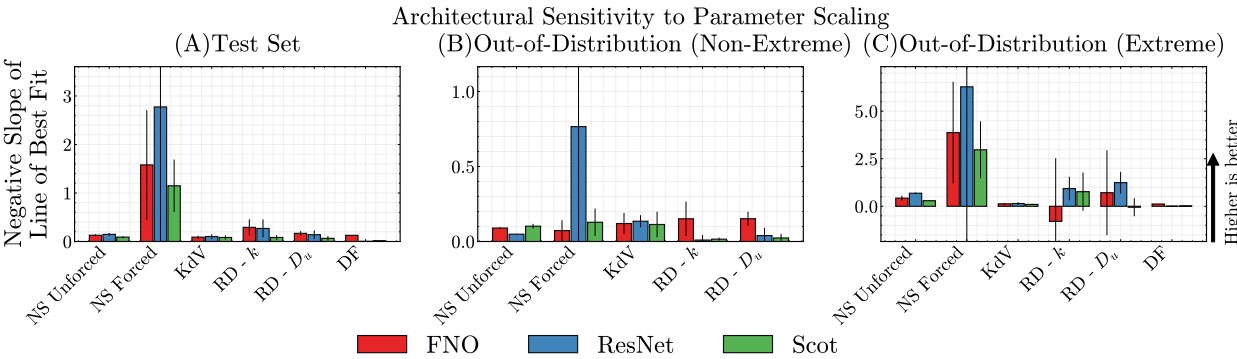

Figure 15: **Effect of PDE parameter scaling across architectures** Evaluating data efficiency of different architectures to scaling the number of PDE parameters. Increasing the total number of initial conditions during training improves performance for all architectures with FNO being the most data efficient.

the temporal horizon during training does not inherently improve generalization capability when estimating PDE parameters from solution fields outside the training distribution.

### D.4 Physicality of Predicted Parameters

While relative error is a convenient way to compare predicted and true parameters, it does not directly test whether the predictions are physically meaningful. A complementary evaluation is to evolve a numerical solution using the predicted parameter and compare physically meaningful diagnostics of the resulting trajectory to the reference trajectory, thus capturing a notion of "self-consistency" of the predictions.

We apply this idea to forced 2D Navier–Stokes, where the viscosity $\nu$ strongly controls the system's dissipation and thus the energy distribution across length scales. In Figure 17 we plot the energy spectra from (i) a reference simulation run with the true PDE parameter, and (ii) a simulation run with the parameter predicted by an FNO-based inverse model when conditioned on frames from the reference simulation. Predictions are shown with orange, solid lines and references with blue, dashed lines. We repeat this for 9 pairs of predicted and true parameters spanning the parameter range.

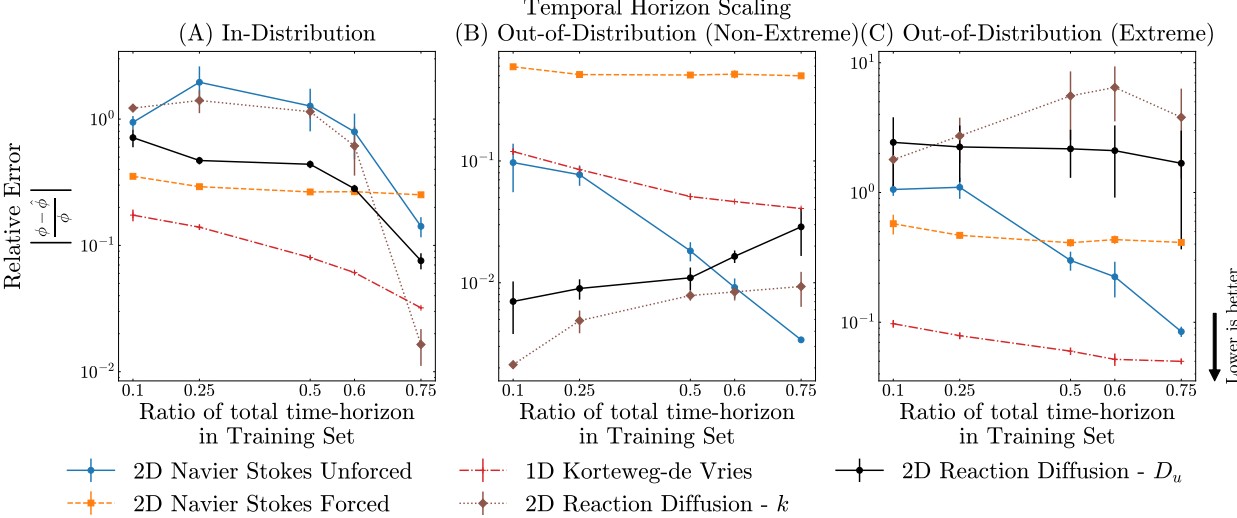

Figure 16: **Effect of increasing the ground truth time horizon.** Evaluating Inverse problem performance on different quantities of available data by scaling the total time horizon of the training solution fields. Increasing the total time horizon of training trajectories improves performance on held out future time frames.

In the reference simulations, more viscous flows (higher $\nu$ / lower Reynolds number) exhibit an earlier drop off in the energy spectrum, whereas lower-viscosity (more turbulent) regimes maintain appreciable energy up to higher wavenumbers, before entering the dissipative range.

Overall, simulations run with the predicted parameters largely preserve the energy spectra. Rolled-out solutions reproduce the qualitative location and shape of the spectral drop-off associated with dissipation. The low-viscosity regime, Figure 17(G-I), is noticeably more sensitive, with small parameter discrepancies leading to visibly larger shifts in the high-wavenumbers, consistent with the turbulent and chaotic nature at low viscosity. For high viscosity parameters, even predictions with higher relative error in Figure 17(A-F) produce physically consistent solutions.

This analysis underscores that evaluation is ultimately task-dependent. Scalar relative errors are useful summary statistics, but physics-based diagnostics such as spectra more directly test whether a predicted parameter leads to the correct qualitative behavior when used in a forward simulator. However, performing this kind of rollout-based, physics-level evaluation exhaustively across all parameters and systems is computationally prohibitive. Even for a single trained model, it would require regenerating a set of simulations comparable in size to the full dataset, multiplied by the number of random seeds (three in our setup), which quickly becomes intractable. We thus recommend that it be used sparingly as a diagnostic tool.

### D.5    Noisy Inputs

We induce partial observability via two degradation operators: salt-and-pepper (S&P) noise and Butterworth (BW) filtering. Salt-and-pepper noise replaces a specified proportion of pixels with either white (salt) or black (pepper) values, modeling random sensor failure or dropout. In contrast, the Butterworth filter removes a fraction of high-frequency modes from the input solution fields, simulating instruments with limited bandwidth or spatial resolution. These two corruption models probe complementary failure modes: S&P evaluates robustness to spatially sparse, unstructured corruption, while BW filtering tests the ability to recover parameters when fine-scale (high-frequency) information is systematically removed.

We construct evaluation splits with varying degradation levels. For S&P noise, we apply corruption levels with probabilities $p \in \{0.2, 0.5, 0.75\}$. For BW filtering, we use a filter of order 6 and remove high-frequency modes with ratios $p \in \{0.2, 0.5, 0.75\}$. We first evaluate models trained on clean data ($p = 0$) to assess their robustness under increasing levels of test-time degradation (Figure 18).

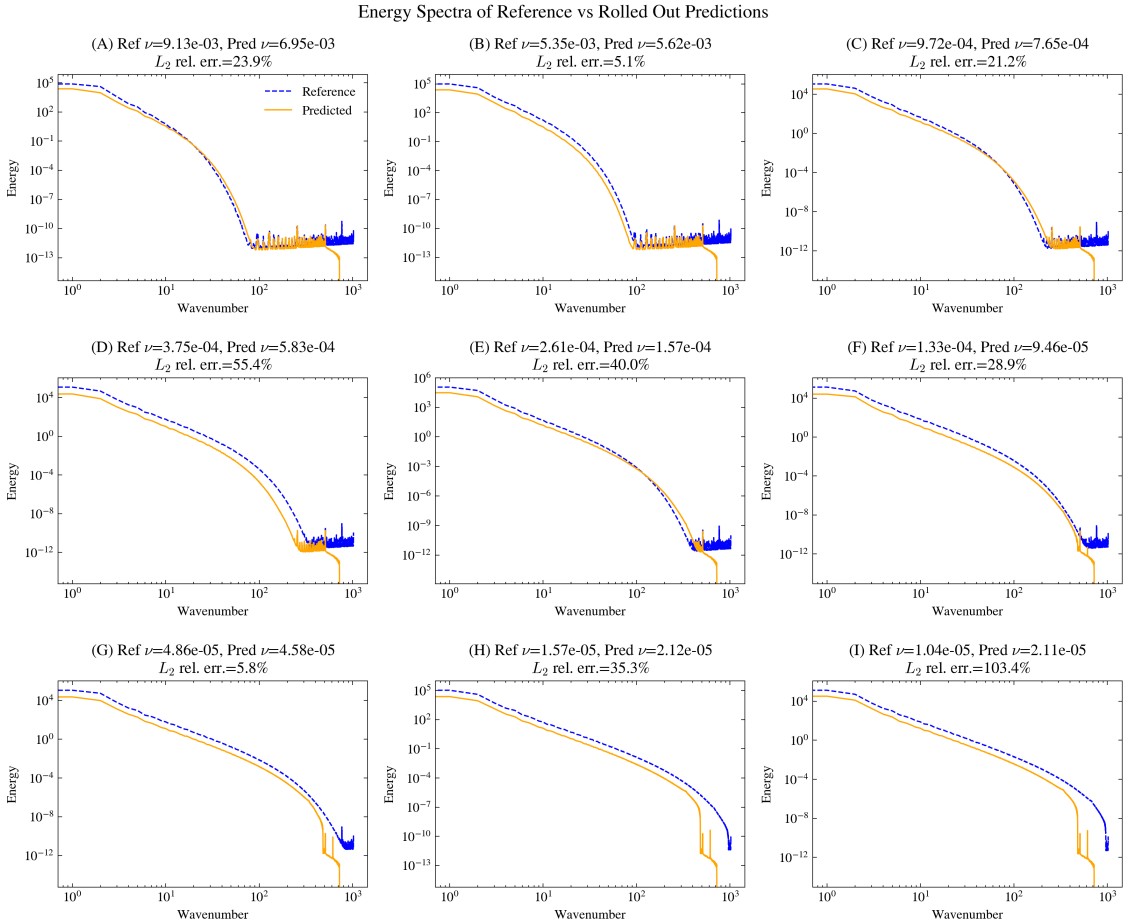

Figure 17: **Energy spectra of predicted Forced 2D Navier Stokes vs. reference solutions.** Energy spectra corresponding to predictions are in solid lines, while references are dashed.

Armed with the insight that removing partial derivative conditioning improves the robustness of the models to degradation (Figure 18), we fix this setting and now train models on varying levels of degradation without conditioning on partial derivatives. In Figure 19, we visualize a heat map of model performance as a function of the level of degradation applied at both train- and test-time.

Across Salt-and-Pepper (SNP) corruption, a clear diagonal preference emerges: models achieve the lowest error when the training corruption level matches the inference corruption level, and performance degrades as one moves away from this match in either direction. In particular, models are generally robust when the inference noise is less than or approximately equal to the training noise, but performance drops more sharply when the inference corruption exceeds what was seen during training. This suggests that the learned inverse operator is effectively calibrated to the corruption distribution encountered during training and does not extrapolate well beyond it. Training with non-zero corruption significantly improves robustness relative to clean-only training, which exhibits the largest degradation under noisy inference. Among the tested settings, training at p=0.5p=0.5p=0.5 provides the most balanced behavior, offering reasonable performance across a range of noise levels without over-specializing. However, this robustness comes at a cost: models trained on corrupted fields consistently underperform on clean inputs compared to models trained purely on clean data.

In contrast, the Butterworth (BW) corruption experiments exhibit qualitatively different behavior. Once models are trained with even moderate filtering (e.g., $p \approx 0.2$), performance becomes largely invariant across the inference corruption axis, with near-constant error across test noise levels. More precisely, training at a given level of degradation yields models that are robust to equal or lower levels of filtering at inference. This suggests that the

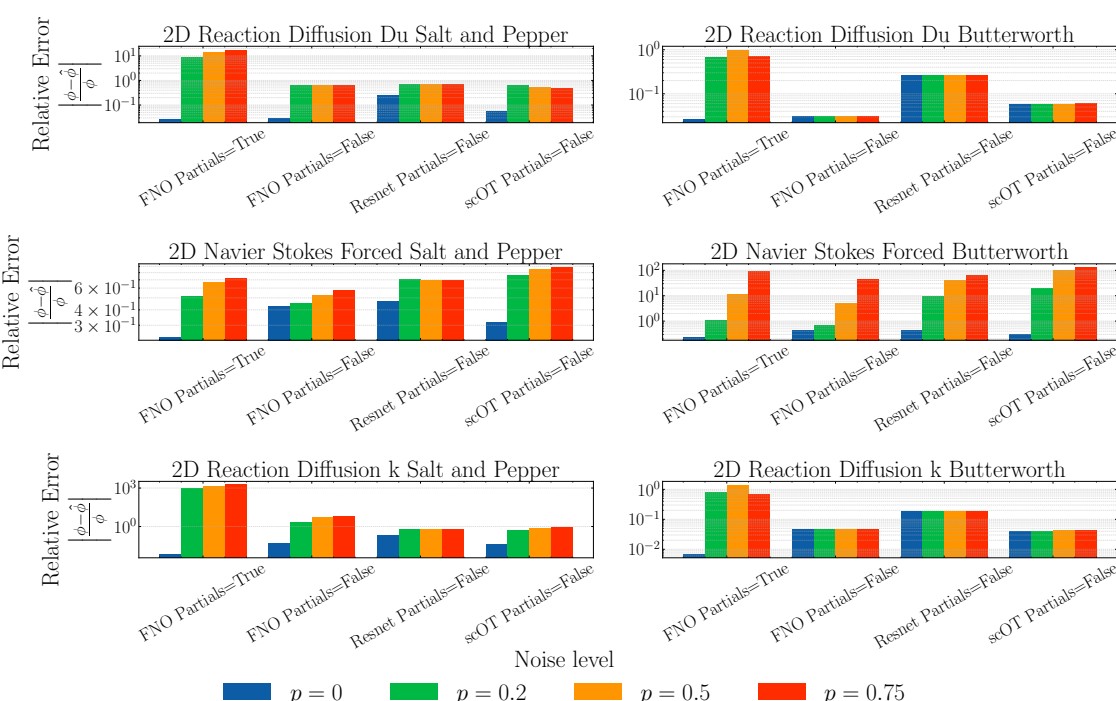

Figure 18: **Impact of noisy inputs on models trained on non-noisy solution fields.** Evaluating models trained on clean inputs across all design axes under varying levels of salt-and-pepper noise and Butterworth filtering. Models trained without partial-derivative inputs exhibit improved robustness to these degradations.

learned inverse mapping is insensitive to the exact magnitude of spectral corruption, provided the corruption class is observed during training. This behavior can be attributed to the smooth, structured nature of Butterworth filtering: the roll-off suppresses high-frequency components while preserving low-frequency structure, effectively constraining the inverse problem to a stable subspace that generalizes across filter strengths.

## D.6 Non-Uniform Grids

We study the impact of non-uniform spatial discretization by randomly removing spatial grid lines with probability $p$, yielding solution fields defined on irregular grids. We evaluate FNO, ResNet, and scOT across increasing levels of grid sparsification, with $p \in \{0.05, 0.15, 0.3\}$ Figure 20. As expected, higher drop probabilities lead to increased reconstruction error; however, the magnitude and trend of degradation vary across models and underlying PDE systems. We hypothesize that ResNet, with it's convolutional bias, performs the worst due to the limited receptive field in early layers, while FNO and scOT can more easily learn global features via the Fourier layers or attention.

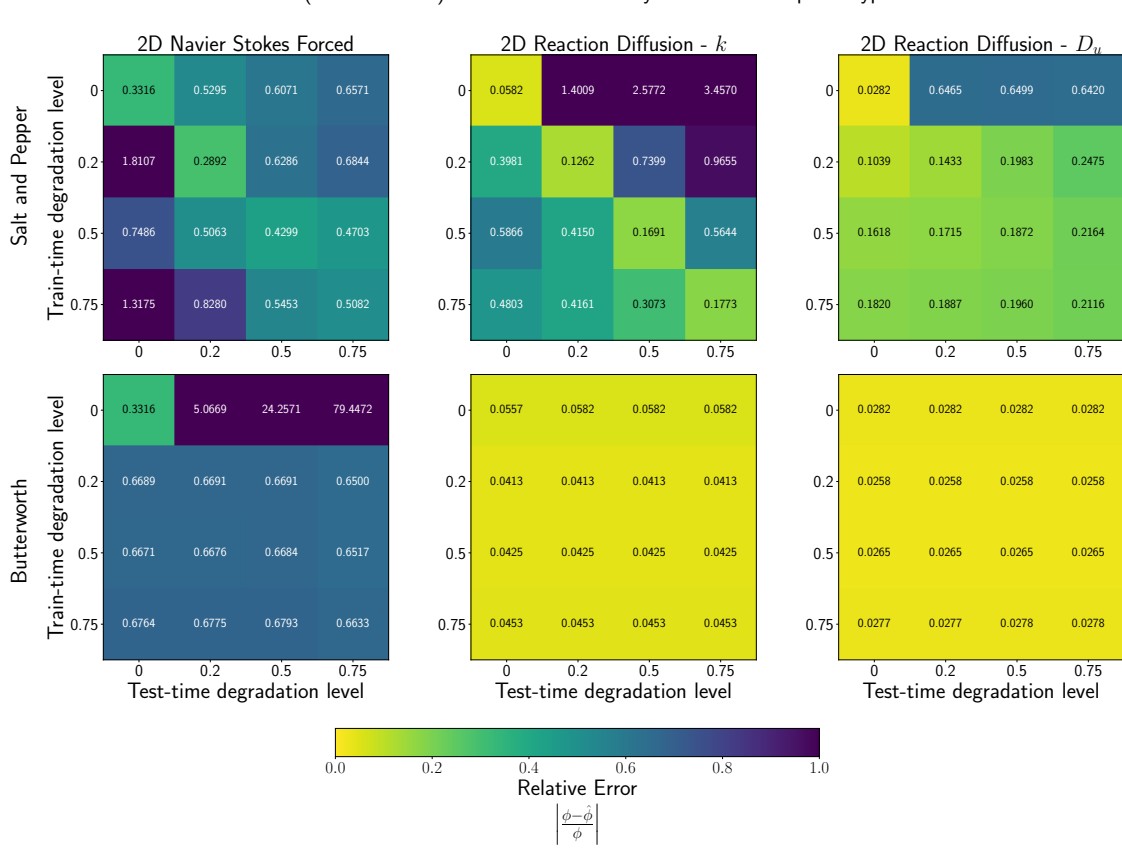

Figure 19: **Evaluation of FNO robustness after training under observational degradation**. Heatmaps show relative error as a function of train-time (rows) and test-time (columns) degradation levels for salt-and-pepper (top) and Butterworth (bottom) corruption across three PDE systems. Salt-and-pepper noise induces a strong diagonal structure, indicating sensitivity to mismatch between training and inference corruption levels, whereas Butterworth filtering yields nearly uniform performance across test-time degradation once seen during training. Training on corrupted inputs improves robustness but introduces a trade-off in performance on clean data.

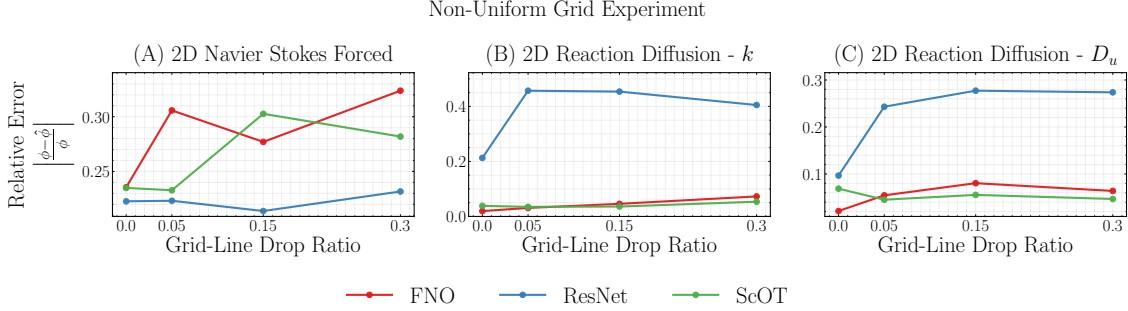

Figure 20: **Impact of Non-Uniform Grids on PDE inverse problem performance.** Comparison of architectural inductive biases (FNO, ResNet, scOT) to varying degrees of non-uniformity in solution fields

## D.7 Benchmark Results

We provide the relative errors of the various models across all systems and settings in a table format.

| System | FNO | ResNet | scOT | DeepONet |
|---|---|---|---|---|
| 2D NS Unforced | 0.0052±0.0004 | 0.0146±0.0008 | 0.0164±0.0015 | 0.0174±0.0010 |
| 2D NS Forced | 0.2402±0.0045 | 0.3964±0.1228 | 0.2589±0.0173 | 0.2032±0.0035 |
| 1D KdV | 0.0147±0.0007 | 0.0197±0.0055 | 0.0223±0.0011 | 0.0166±0.0006 |
| 2D RD $-k$ | 0.0331±0.0110 | 0.3426±0.0968 | 0.0453±0.0049 | 0.4129±0.0194 |
| 2D RD $-D_u$ | 0.0229±0.0020 | 0.1740±0.0559 | 0.0749±0.0049 | 0.2406±0.0306 |
| 2D DF | 0.0671±0.0032 | 0.0006±0.0001 | 0.0181±0.0005 | 0.0006±0.0003 |

Table 6: Model relative error on the test set. The best value in each row is underlined.

| System | FNO | ResNet | scOT | DeepONet |
|---|---|---|---|---|
| 2D NS Unforced | 0.0039±0.0002 | 0.0086±0.0005 | 0.0320±0.0046 | 0.0097±0.0003 |
| 2D NS Forced | 0.5026±0.0242 | 0.2390±0.0038 | 0.4657±0.0550 | 0.2331±0.0215 |
| 1D KdV | 0.0362±0.0021 | 0.0635±0.0054 | 0.0472±0.0008 | 0.0624±0.0022 |
| 2D RD $-k$ | 0.0126±0.0025 | 0.2737±0.0508 | 0.0249±0.0048 | 0.2551±0.0863 |
| 2D RD $-D_u$ | 0.0292±0.0042 | 0.1285±0.0280 | 0.0991±0.0310 | 0.2041±0.0727 |

Table 7: Model relative error on OOD (Non-Extreme) set. The best value in each row is underlined.

| System | FNO | ResNet | scOT | DeepONet |
|---|---|---|---|---|
| 2D NS Unforced | 0.0831±0.0039 | 0.1608±0.0128 | 0.0679±0.0046 | 0.1919±0.0433 |
| 2D NS Forced | 0.3998±0.0208 | 0.9751±0.3837 | 0.3669±0.0715 | 0.5144±0.0149 |
| 1D KdV | 0.0465±0.0018 | 0.0758±0.0245 | 0.0569±0.0028 | 0.0546±0.0024 |
| 2D RD $-k$ | 11.0376±0.5903 | 1.8791±0.4135 | 2.3177±0.1775 | 1.8405±0.4136 |
| 2D RD $-D_u$ | 2.2911±1.2060 | 2.5445±0.3617 | 1.6681±0.3646 | 2.2492±0.6320 |

Table 8: Model relative error on the OOD (Extreme) set. The best value in each row is underlined.

