### D.3.3 Effect of data scaling across architectures

We investigate how the performance of each model architecture behaves when scaling data along initial conditions and PDE parameters.

**Effect of scaling initial conditions across architectures** Using more initial conditions generally leads to better performance for all architectures in most evaluation settings and systems, evidenced by Figure 17. The relative ordering of architectures under initial-condition scaling is largely consistent across in-distribution and out-of-distribution evaluation regimes, with FNO consistently exhibiting the strongest gains from increased initial-condition diversity. This potentially suggests that the inductive biases in FNO benefit more strongly from increased number of initial conditions, compared to other architectures.

**Effect of scaling PDE parameters across architectures** More PDE parameters generally leads to better performance for all architectures in most evaluation settings and systems (Figure 18). Unlike initial-condition scaling, the performance improvements and the relative ordering of architectures under parameter scaling

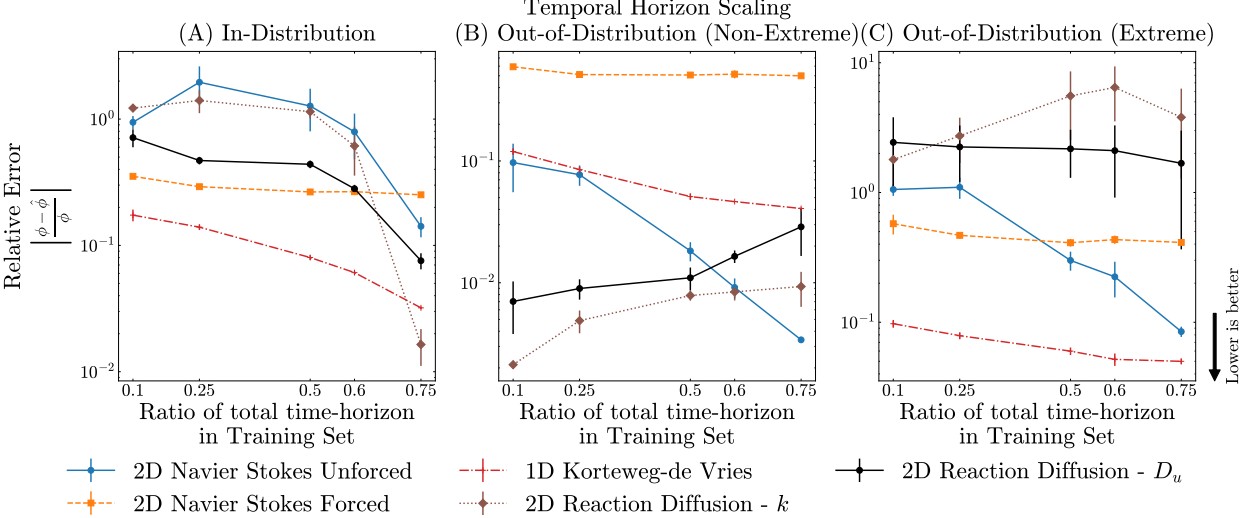