# OpenReview forum: "PDEInvBench: A Comprehensive Dataset and Design Space Exploration of Neural Networks for PDE Inverse Problems"
_TMLR — Accepted by TMLR_

### Review · Reviewer_rvMi · 2026-02-24

**Summary Of Contributions:**

This paper presents a comprehensive benchmark PDEInvBench for the PDE inverse problem, which includes both 1D and 2D, time-dependent and time-independent inverse tasks. I think this work can be a good supplement to the current forward-modeling-dominating research. Besides, the authors also provide a detailed comparison and analysis of three types of model architectures and scaling properties. The proposed key insights are inspiring.

**Audience:**

Yes

**Audience Explanation:**

This paper can be a good supplement to current research, which can also be used in benchmarking neural operators.

**Claims And Evidence:**

Yes

**Claims Explanation:**

As a benchmark paper, I think the authors have provided a sufficient description of their design principle and experimental details.

**Requested Changes:**

More experiments about irregular mesh and partial observation should be added.

Although the current experiments are sufficient for a pure technical paper, I believe the authors should also include PDEs in 3D domains, irregular meshes, and partial observation, since the above-mentioned scenarios are the ones that require system identification. Note that one mainstreaming trend of PDE solving architecture is tackling the irregular mesh, where some advanced models have been proposed, e.g., Transolver or UPT. Due to the scope limitation of the current benchmark, none of these recent irregular-mesh models has been tested.

I appreciate that such a limitation has also been discussed in the paper. If the authors can further enlarge the benchmark scope, this paper will be more likely to be impactful.

---

> ### Author Response · Authors · 2026-04-02
> **Response to Reviewer rvMi**
>
> We thank the reviewer for their comments. We have added new experiments to address their concerns regarding partial observability and irregular meshes.
>
> ### Partial Observability
> We induce partial observability via two degradation operators: salt and pepper noise, and a Butterworth filter. The former replaces a specified proportion of pixels with a purely white (salt) or pepper (black) pixel, while the latter drops a certain proportion of high-frequency modes from the input solution fields. These are noise models for realistic inverse problems representing random sensor failure / dropout (salt and pepper) or limited sensor bandwidth (Butterworth filter). The former tests robustness to spatially sparse corruption while the latter tests the models’ ability identify parameters when high-frequency details are missing. The full experimental setup and results are in Appendix F.
>
> We create several new evaluation splits corresponding to varying proportions of degradation from $p = 0.2$ to $p=1$. We first take the best performing models along our design axes, which were trained on uncorrupted data ($p = 0$), and evaluate their robustness to varying amounts of degradation at test-time, visualized in Appendix Figure 21.
>
> Interestingly, we find that that removing partial derivative conditioning improves the robustness of the models to degradation. We then fix this setting and now _train_ models on varying levels of degradation. A heat map visualizing model performance as a function of the level of degradation applied at both train- and test-time is shown in Appendix Figure 22.
>
> For salt-and-pepper noise, the performance is sensitive to the mismatch between the train-time and test-time corruption levels, with the best results concentrated along the diagonal. While training on corrupted inputs degrades absolute performance on clean data, it does improve robustness at higher test-time degradation levels. For Butterworth filtering, any degradation is useful for improving the model’s robustness to noise. The larger performance degradation under the Butterworth filter for forced Navier-Stokes compared to reaction-diffusion is consistent with turbulent flows relying more heavily on high-frequency spatial content for parameter identification, while the smoother dynamics of reaction-diffusion are impacted less by bandwidth limitation.
>
> ### Irregular Meshes
> We agree that irregular mesh architectures are in important direction in PDE modeling. However, to our understanding, architectures like Transolver and UPT operate on point cloud representations, which would require rethinking our input pipeline compared to the current grid-based methods in our benchmark. Integrating them would require a careful design of data representation, training infrastructure, and evaluation protocol. We would like to investigate such approaches in future follow-up work, where we can dedicate the resources for a fair and insightful comparison.
>
> As a middle ground for the current paper, we investigate the effect of non-uniform spatial grids by dropping spatial lines from the solution field, creating grids with variable spacing. This tests a slightly different question: how do existing grid-based architectures degrade when the assumption of uniform spacing is violated? We believe this is more informative than adding a single irregular-mesh architecture in isolation since it contextualizes the robustness of the models already under study rather than introducing a new architecture without careful controlled comparison.
>
> We do acknowledge that non-uniform rectilinear grids are not the same as truly unstructured meshes (e.g., triangular), and extending the benchmark to support unstructured mesh representations is a valuable direction for future work.
>
> We visualize the results in Appendix G Figure 23. In general, we witness a degradation in performance. We hypothesize that ResNet, with its convolutional layers, performs the worst due to the limited receptive field in early layers. The other two architectures, scOT and FNO, propagate global attention either through the Fourier layers or attention, making them more robust to dropped grid points.
>
> ### 3D Domains
> Generating, storing, and training on 3D PDE datasets at the scale required for a comprehensive benchmark study would be computationally prohibitive. The simulation cost alone scales cubically with spatial resolution, and the resulting datasets would require substantially more storage and GPU memory for training, making the controlled multi-axis design exploration we perform infeasible at the 3D scale with current resources. While this is an exciting direction, we leave this extension to future work.

---

> > ### Comment · Reviewer_rvMi · 2026-04-12
> >
> > Thanks for the author's response. My concerns were completely resolved by the rebuttal. Thus, I kept my acceptance recommendation.

---

### Review · Reviewer_FZki · 2026-03-03

**Summary Of Contributions:**

Authors propose a benchmark for inverse problems that includes identification of parameters in $4$ PDEs: $D=2+1$ reaction-diffusion, $D=2+1$ Navier-Stokes (with and without forcing), $D=1+1$ Korteweg-De Vries, $D=2$ stationary diffusion equation. In addition to dataset authors develop a special evaluation scheme to assess generalisation in and out of distribution.

The results on proposed datasets are reported for three distinct classes of architectures (convolution, neural operator, transformer) trained with residual, $L_2$ and anchor loss (for finetuning). The discussion of results contains a list of concise actionable observations.

**Audience:**

Yes

**Audience Explanation:**

Inverse problem is a field on the intersection of several research areas including generative modelling, scientific computing (including scientific computing done with ML), regularisation techniques, MCMC, etc. So the topic itself is of wide interest.

As authors rightfully noted, benchmarks for forward scientific problems (for PDEs) are more abundant than benchmarks for inverse problems. Some specific inverse problems in computer vision (e.g., inpainting) are well covered in the literature, but scientific inverse problems provide a more diverse scenario and often more challenging problems. Given that, the specific benchmark and findings of authors is a valuable addition to the scientific literature and can potentially be of interest for many individuals in TMLR's audience.

**Claims And Evidence:**

Yes

**Claims Explanation:**

I believe many claims of the authors are supported by clear evidence. The only major unsupported claim is the novelty of the benchmark. This is further explained below.

**Requested Changes:**

**Missing literature**

In my view authors slightly downplay the significance of benchmarks for inverse problems that already exist in literature. I would advise authors to reconsider they position and discuss at least the following related works:

1. InverseBench: Benchmarking Plug-and-Play Diffusion Priors for Inverse Problems in Physical Sciences, https://arxiv.org/abs/2503.11043.

   ICLR 2025 spotlight covers classical Full Waveform Inversion, Linear Inverse Scattering and reconstruction of initial conditions for the Navier-Stokes equation.

2. A benchmark for the Bayesian inversion of coefficients in partial differential equations, https://arxiv.org/abs/2102.07263.

   The problem authors consider is identification of diffusion coefficient for stationary diffusion equation (Darcy flow).

3. PDEBENCH: An Extensive Benchmark for Scientific Machine Learning, https://arxiv.org/abs/2210.07182

   Authors discussed PDEBENCH but claim that "evaluates several PDE systems but only generates a few PDE parameters per system, making it ill-equipped to evaluate the performance of inverse models over a wide range of physical behaviors". I find this description is debatable. In said benchmark the inverse problem was to identify initial conditions based on observations made at $T > 0$. Such kinds of inverse problems can also be challenging and useful in many scenarios, e.g., data assimilation in numerical weather forecasts.

4. The Well: a Large-Scale Collection of Diverse Physics Simulations for Machine Learning, https://arxiv.org/abs/2412.00568

   The Well is also discussed with the description "a recent large dataset for the forward problem, covers a broad set of PDEs and focuses on generating a large number of initial conditions, but only covers a small range of physical parameter values". I do not find this description accurate: (i) two datasets (`acoustic_scattering` and `helmholtz_staircase`) in the Well are specifically designed for inverse problems; (ii) an interesting way to use The Well dataset is to withhold information about parameters from the model and force it to make prediction from a short trajectory (see https://arxiv.org/abs/2409.02313), this way the model implicitly solve inverse problem.

5. Neural Inverse Operators for Solving PDE Inverse Problems, https://arxiv.org/abs/2301.11167.

   Several inverse problems are considered: Calderon Problem, Inverse Wave Scattering, Seismic Imaging, etc.
I do not think the existence of other benchmarks somehow invalidate the research done by authors, but in my view a more extensive discussion of the existing literature will strengthen the submission.

**Problem setup and evaluation**

One notable challenge in inverse problems is they may be ill-posed, meaning many parameters may describe the same of very close observations. The setup authors proposed does not take this problem into account.

More formally, suppose I have a forward model $y = f(x)$ with input $x$ and output $y$. I have observations of the output $o = \Pi(y)$ and the goal is to identify $x$. It may well be the case that one can find $x_1$ and $x_2$ such that $x_1 \neq x_2$ and $\Pi\left(f(x_1)\right) \simeq \Pi\left(f(x_2)\right)$ or even $f(x_1)\simeq f(x_2)$. If the ground truth data is $x_1$ and my model predicts $x_2$ the relative error $\left\|x_2 - x_1\right\|\big/\left\|x_1\right\|$ may be large but the model is working fine. Can the authors discuss this problem and explain how they avoid it?

In particular, it would be great to see the following additional investigations:
1. Estimation of the sensitivity of the forward model to the inferred parameters, e.g., singular values of $\frac{\partial f(x)}{\partial x}$.

2. Study of the relative forward error $\left\|f(x_2) - f(x_1)\right\|\big/\left\|f(x_1)\right\|$ for the inferred parameter $x_2$ and ground truth parameter $x_1$.

**Minor issues**

1. Equation (4) is referred to as "self-supervised loss" in several places in the article. To compute this for non-stationary equations one still needs to know the part of the trajectory since the neural network uses this information to predict parameters of the PDE. To me this looks like a standard supervised problem but with residual loss. Can the authors explain why the describe this loss as "self-supervised"?

2. Can the authors explain the meaning of column "Number of Parameter Values" in Table 1? For example, for the KdV equation we have a single parameter $\delta$ but we have $100$ in the mentioned columns. Does it mean the data was generated for $100$ distinct values of $\delta$ on the uniform grid between $0.8$ and $5$? Can the author provide explanations on all equations listed in Table 1.

3. On page 8 authors claim "Both FNO and scOT are by construction neural operators, and we include both to examine different classes of models. While not strictly a neural operator, ResNet operates over arbitrarily sized inputs due to the nature of convolutions which makes it a useful point of comparison. However, note that it is not a discretization invariant like FNO or scOT." But from the appendix it is evident that FNO is used only as an encoder that is later complemented with a series of standard convolutions with downsampling. Given that, the resulting architecture is clearly not "discretisation invariant". Can the authors comment on that and also on the discretisation invariance of scOT? In general, why do authors believe the "discretisation invariance" is important for the problem at hand?

4. One of the conclusions by the authors (page 11) " In other words, in our setting, new initial conditions provide a better demonstration of the underlying mapping between solution fields and parameters." In the author's view is it possible to obtain a similar result for Darcy flow? For example, by varying the source or forcing term $f(x)$ in $-\text{div} \left(k(x) \text{ grad } u(x)\right) = f(x)$.

---

> ### Author Response · Authors · 2026-04-02
> **Response to Reviewer FZki (Literature)**
>
> We thank the reviewer for their insightful comments. We split our response across two comments, with this focusing on background literature pointed out by the reviewer..
>
> ### Missing Literature
> We thank the reviewer for pointing out existing benchmarks for inverse problems. We have incorporated the suggested references into the manuscript. Below we highlight some of the key comparisons:
>
> > 1. InverseBench: Benchmarking Plug-and-Play Diffusion Priors for Inverse Problems in Physical Sciences
>
> This paper focuses on diffusion models for solving inverse problems - it  focuses on a broad class of inverse problems including but not limited to the PDE domain. As a result, they do not consider certain PDE-specific design choices, such as the use of PDE residual losses, test-time tailoring, and PDE-specific architectures like FNO. Additionally, their evaluation is performed over multiple initial conditions for a fixed PDE parameter (e.g. Re = 200), rather than over both multiple ICs/params like our work.
>
> > 2. A benchmark for the Bayesian inversion of coefficients in partial differential equations
>
> In this paper, the authors introduce an inverse benchmark based on the Poisson equation. A rigorous Bayesian formalism is developed for estimating the posterior distribution of the PDE parameter, but neural networks are not involved, with samples instead generated with MCMC.
>
> > 3. PDEBENCH: An Extensive Benchmark for Scientific Machine Learning
>
> Our intended point is simply that PDEBENCH contains data generated at one or a few unique PDE parameters, while varying the initial conditions. Meanwhile, our work generates data with both multiple initial conditions and a range of unique PDE parameters. We agree that the initial condition prediction flavor of inverse problem is also useful, so we have removed the statement regarding PDEBENCH being “ill-equipped to evaluate the performance of inverse models over a wide range of physical behaviors.”
>
> > 4. The Well: a Large-Scale Collection of Diverse Physics Simulations for Machine Learning
>
> We thank the reviewer for the correction and have since revised our description of The Well in the manuscript. We acknowledge that several datasets, including the mentioned datasets `acoustic_scattering` and `helmholtz_staircase` , could be used for inverse problems and that The Well covers a broad range of physical parameter values in its datasets.
>
> Our work differs in our evaluation focus and analysis. The Well’s baselines and benchmarks target the forward problem, noting that inverse problems as a future direction (section 4.2, Future Challenges [1], “While our example baselines target the forward problem, … Several datasets, such as `acoustic_scattering` and `helmholtz_staircase` are well-suited for inverse scattering tasks”). PDEInvBench focuses explicitly on inverse problems in both evaluations and analysis done. We believe the two efforts are complementary, and that integrating datasets from The Well into our inverse problem evaluation framework is a natural direction for future collaborations.
>
> > 5. Neural Inverse Operators for Solving PDE Inverse Problems
>
> We have expanded our discussion of [2] in the revised manuscript. Neural Inverse Operators (NIO) is proposed as a combination of DeepONet and FNO for recovering spatial coefficient fields from boundary measurements. Our work differs in that we focus on benchmarking and design space analysis that are applicable across architectures. PDEInvBench and NIO also cover complimentary PDE systems. It would be interesting to apply our findings to NIO, but this is left as future work.
>
>
> [1] Ruben Ohana, Michael McCabe, Lucas Meyer, Rudy Morel, Fruzsina J. Agocs, Miguel Beneitez, Marsha Berger, Blakesley Burkhart, Keaton Burns, Stuart B. Dalziel, Drummond B. Fielding, Daniel Fortunato, Jared A. Goldberg, Keiya Hirashima, Yan-Fei Jiang, Rich R. Kerswell, Suryanarayana Maddu, Jonah Miller, Payel Mukhopadhyay, Stefan S. Nixon, Jeff Shen, Romain Watteaux, Bruno Régaldo-Saint Blancard, François Rozet, Liam H. Parker, Miles Cranmer, & Shirley Ho. "The Well: A large-scale dataset collection for scientific machine learning," in 38th Conference on Neural Information Processing Systems (NeurIPS 2024) Track on Datasets and Benchmarks, 2024.
>
> [2] R. Molinaro, Y. Yang, B. Engquist, and S. Mishra. Neural Inverse Operators for Solving PDE Inverse Problems. In A. Krause, E. Brunskill, K. Cho, B. Engelhardt, S. Sabato, and J. Scarlett, editors, Proceedings of the 40th International Conference on Machine Learning, volume 202 of Proceedings of Machine Learning Research, pages 25105–25139. PMLR, July 2023. URL https://proceedings.mlr.press/v202/molinaro23a.html.

---

> ### Author Response · Authors · 2026-04-02
> **Response 2 to Reviewer FZki (Problem Setup)**
>
> We address comments about the problem setup here.
>
> ### Problem setup and evaluation
> We thank the reviewer for their comments and for raising this important point. Ill-posedness is an important challenge in inverse problems, and we have updated the manuscript to mention this. We now discuss how our benchmark addresses it.
>
> We also argue that our dataset generally does not contain a significant portion of degenerate cases, where different parameters map to the same solution field. As a demonstration of this, we include an animation (https://zenodo.org/records/19244371, download to animate) showing that the energy spectrum of forced 2D Navier Stokes varies smoothly and monotonically as a function of viscosity, indicating that solutions themselves will also be similarly well-behaved/non-degenerate.
>
> In Appendix D.4 we provide a forward-error analysis for forced 2D Navier-Stokes. The idea is to evolve a numerical simulation using the predicted parameter and compare physically meaningful diagnostics of the resulting trajectory to a reference trajectory with the correct value of the parameter, thus capturing a notion of “self-consistency" of the inverse model predictions w.r.t the forward error.
>
> Specifically, we compare the energy spectra of forward simulations run with the predicted and ground-truth viscosity parameters. The energy spectra captures where solutions agree or diverge in wavenumber space. We find that the predicted parameters produce physically consistent energy spectra across the parameter range, with discrepancies concentred in the high-wavenumber range for the most turbulent (low viscosity) regime. We have focused this analysis on the forced Navier-Stokes case because it represents a challenging setting in our benchmark, and the chaotic dynamics of the forward map make the observed solution field sensitive to the parameter perturbations, reflecting a scenario where concerns of ill-posedness are prominent.
>
> As the reviewer mentions, the identifiability of a inverse problem depends not only on the forward map $ f $ but also on the observation operators $ \Pi $, which in our setup comprises a mix of: (i) temporal subsampling (observing only $ n_\text{past} $ frames from the trajectory), (ii) finite spatial resolution, and (iii) the optional inclusion of partial derivative channels. Each of these affects the information available for the inverse problem, and we investigate these directly in Section 5 and Appendix D.2. Our experiments systematically study how the choice of observation operator affects identifiability in an ill-posed setting. In the rebuttal to reviewer rvMi, we also include results with partial/noisy observability.
>
> In this work, for simplicity of training and benchmark evaluation, we have adopted a discriminative approach to the inverse problem. To better capture ill-posedness or uncertainty, a generative framework should be employed. This would allow us to capture the full posterior distribution of parameters $ p(x | y) $ given the observed solution field $ y $. We plan to pursue this in future work.

---

> ### Author Response · Authors · 2026-04-02
> **Response to Reviewer FZki (Minor issues)**
>
> We address the minor issue comments provided by the reviewer.
>
> ### Minor Issues
> > 1.  Can the authors explain why the describe this [Eqn. (4)] loss as "self-supervised"?
>
> We describe this loss as self-supervised because it doesn’t require knowledge of the prediction target (i.e., the true PDE parameter). Recall from Eq. 4 that the residual loss is defined as $$ L_{\text{res}} = \|F_{\hat{\phi}}(u_{t - k}^i, ..., u_t^i)\|^2_2 = \|F_{f_\theta(u_{t - k}^i, ..., u_t^i)}(u_{t - k}^i, ..., u_t^i)\|^2_2 $$. Nowhere in this loss does the true PDE parameter $ \phi $ appear. Meanwhile, the standard supervised data loss (Eq. 3) is defined as $ L_\text{data} = \frac{\| f_\theta(u_{t - k}^i, ..., u_t^i) - \phi^i\|_2}{||\phi^i||_2} $. Computing this loss does require knowledge of the true PDE parameter.
>
> > 2. Can the authors explain the meaning of column "Number of Parameter Values" in Table 1?
>
> The “Number of Parameter Values” column indicates the number of distinct parameters sampled either linear-uniformly or log-uniformly from the range specified in the “Parameters” column, that are included in the dataset (union of training and all evaluation splits). We have also added in parenthesis the sampling scheme that was used for each PDE. For example, for 2D Navier-Stokes (forced), we use 120 distinct values of $ \nu $ sampled log-uniformly between $ \nu \in [10^{-5}, 10^{-2}] $. We have clarified this in the manuscript.
>
> > 3. In general, why do authors believe the "discretisation invariance" is important for the problem at hand?
>
> The reviewer raises a fair point. The full inverse-problem pipeline we are using - an encoder followed by a convolutional downsampling and an MLP regression head - is not discretization invariant, and we have made this clear in the revised manuscript. For inverse problems targeting a scalar parameter, strict discretization invariance is impossible since any architecture must ultimately reduce a spatial field to a single number which requires some form of pooling or downsampling.
>
> However, the encoder itself, which is the component where the architectural inductive bias is most consequential, does retain the discretization invariance. Our motivation for using these architectures is not to exploit this property end-to-end, but stems from the fact that these architectures were developed for PDE applications and their inductive biases are beneficial for processing solution fields. We are interested in whether these inductive biases yield advantages for inverse problems as well, like the forward problem. We have correspondingly softened the claims of discretization invariance in the paper.
>
> > 4. In the author's view is it possible to obtain a similar result for Darcy flow?
>
> Our current dataset uses $f(x) = 1$, following [1]. We expect the scaling benefit from varying $f(x)$ to be significantly more modest compared to scaling initial conditions in the nonlinear, time-dependent case. This is because Darcy Flow is a linear elliptic PDE; solutions corresponding to different source terms are related by superposition, so each new $f(x)$ provides a linearly dependent view of the permeability. This is in contrast to the nonlinear time-dependent systems we study where different initial conditions can induce unique physical dynamics.
>
> [1] Z. Li, N. B. Kovachki, K. Azizzadenesheli, B. liu, K. Bhattacharya, A. Stuart, and A. Anandkumar. Fourier Neural Operator for Parametric Partial Differential Equations. In International Conference on Learning Representations, 2021. URL https://openreview.net/forum?id=c8P9NQVtmnO.

---

> > ### Comment · Reviewer_FZki · 2026-04-12
> >
> > I would like to thank the authors for a detailed reply and for incorporating many changes into the manuscript. I especially appreciate the additional experiments that evaluate simulation error with inferred parameters for the Navier-Stokes equation. Although the authors advised against performing such an evaluation in a general setting ("be used sparingly as a diagnostic tool"), I am positive that case-by-case analysis motivated by physics is preferable to reducing evaluation to a single metric, such as accuracy.
> >
> > My primary concerns were successfully resolved by the rebuttal, so I recommend accepting the paper to TMLR.

---

### Review · Reviewer_Xivm · 2026-03-13

**Summary Of Contributions:**

**Brief Summary:** The paper introduces the benchmark `PDEInvBench` dataset for evaluating ML approaches to PDE inverse problems, i.e. the task of estimating parameters (eg. viscosity, diffusivity, etc..) from observed solutions. The dataset covers 5 PDE systems, namely: 2D Reaction Diffusion, unforced + forced 2D Navier-Stokes, 1D Korteweg-De Vries, and 2D Darcy Flow.

The range of parameters is chosen to span qualitatively different physical regimes. Furthermore, the evaluation protocol includes in-distribution, out-of-distribution non-extreme, and out-of-distribution extreme.

The authors conduct a systematic exploration of the design space along 3 axes:
1. optimization procedure, i.e. supervised vs. self-supervised vs. test-time training
2. problem representation (FNO / ResNet / scOT architectures / etc..)
3. scaling with respect to model size, initial conditions, parameter coverage, temporal horizon

The authors argue that: a two-stage training recipe comprised of supervised pretraining + test-time PDE-residual fine-tuning works best; derivative conditioning does help; FNO methods generally outperforms competing approaches on time-dependent PDEs; scaling initial conditions matters more than scaling parameter coverage.

**Strengths**: The paper is timely and fills an important gap in the scientific ML benchmarks (comprehensive benchmarks do exist for forward PDE but not for inverse problems with wide range of parameters). The identified three-axis investigation is well-structured and provides actionable insights/guidance for practitioners.

The experimental methodology is sound (eg. 3 seeds + error bars + train/test splits).
Dataset and code are publicly released.

**Weaknesses**:
1. Classical optimization baselines are weakly tuned/configured.
2. Experiments use noise-free observations, which limits practical applicability & conclusions.

**Audience:**

Yes

**Audience Explanation:**

Several sub-communities would benefit from this work:
1. the neural operator community would gain from a standardized inverse problem benchmark dataset
2. practitioners working on PDE parameter estimation
3. the broader scientific ML community

The benchmark fills a genuine gap -- I am not aware of similar inverse problem dataset/benchmark and I can definitely see myself using the proposed dataset. Furthermore. the paper's practical recommendations are directly usable.

**Broader Impact Concerns:**

NA.

**Claims And Evidence:**

Yes

**Claims Explanation:**

**Claim 1 (Section 6.1): Supervised data-driven training consistently outperforms self-supervised PDE residual training.**:
Figure 2 A/B/C provides strong evidence across all systems

**Claim 2 (Section 6.1): Test-time training (TTT) improves performance**:
Figures 2 D/E and the per-system breakdowns in Figure 10 (Appendix D.1.5)

**Claim 3 (Section 7.1): FNO outperforms ResNet and scOT for time-dependent PDEs**:
Figure 3 A/B/C. *but* the three architectures differ in more than their inductive bias: they have different normalization schemes / skip connection patterns / depth / ratio of local to global operations. The paper acknowledges this.

**Claim 4 (Section 6.2): Derivative conditioning consistently improves performance**:
Figure 3 D/E/F shows clear improvement across systems.

**Claim 5 (Key Insight 4, Section 6.3): Scaling initial conditions yields greater performance gains than scaling PDE parameters.**
Figure 4 D/E/F supports this claim for FNO.

**Requested Changes:**

1. The author attributes FNO advantage to its "spectral inductive bias". The current evidence shows FNO performs best, but does not isolate the cause, I think. The architectures differ in normalization, skip connections, activation functions, receptive field, not only inductive bias. So it may be worth to word this claim slightly differently.

2.  It may be useful to strengthen the classical optimization baselines. Table 5 (Appendix D.1.1) uses the mean parameter as initial guess on a non-convex landscape; poor convergence is expected and does not demonstrate that neural approaches are fundamentally better. Would it be computationally doable to run each classical method with multiple random initializations (e.g., 10 restarts drawn uniformly from the parameter range). If classical methods still underperform, then this would be more convincing.

3. I think it is worth discussing somewhere in the paper that in many applied scenarios, observations are contaminated by noise.

4. I believe it would make a lot of sense to include DeepONet-basedbaselines as it is an extremely popular approach. This would broaden the architectural comparison beyond FNO, ResNet, and scOT. This would also strengthen the paper's claim of being a "comprehensive" benchmark.

5. A few words to better justify the NLS metric would likely strengthen the text.

---

> ### Author Response · Authors · 2026-04-02
> **Response to Reviewer Xivm**
>
> We thank the reviewer for their thoughtful comments and have made the requested changes which we detail below.
>
> ### 1. FNO's Spectral Advantage
> >The author attributes FNO advantage to its "spectral inductive bias". The current evidence shows FNO performs best, but does not isolate the cause, I think. The architectures differ in normalization, skip connections, activation functions, receptive field, not only inductive bias. So it may be worth to word this claim slightly differently.
>
> We agree and have softened this claim in the revised manuscript. We note in Section 5 that the architectures differ in normalization, skip connections, depth, and other factors beyond their primary inductive bias. We have revised the wording to make it clear that FNO’s strong performance is consistent with the idea that a spectral bias is useful, without attributing causality.
>
> ### 2. Classical Baselines
> >It may be useful to strengthen the classical optimization baselines. Table 5 (Appendix D.1.1) uses the mean parameter as initial guess on a non-convex landscape; poor convergence is expected and does not demonstrate that neural approaches are fundamentally better. Would it be computationally doable to run each classical method with multiple random initializations (e.g., 10 restarts drawn uniformly from the parameter range). If classical methods still underperform, then this would be more convincing.
>
> We have updated the Appendix D.1.1 with the classical optimization baselines averaged over 10 initial conditions drawn uniformly from the parameter range. We observe similar results from before, supporting the claim that classical baselines are considerably weaker than ML methods. Appendix table 5 has been updated to use 10 initial condition seeds.
>
> ### 3. Noisy Observations
> >I think it is worth discussing somewhere in the paper that in many applied scenarios, observations are contaminated by noise.
>
> Please see our response to Reviewer rvMi, in which we have included several new experiments related to partial observability from added noise.
>
> ### 4. DeepONet Baseline
> > I believe it would make a lot of sense to include DeepONet-basedbaselines as it is an extremely popular approach. This would broaden the architectural comparison beyond FNO, ResNet, and scOT. This would also strengthen the paper's claim of being a "comprehensive" benchmark.
>
> We agree that DeepONet is an important architectural paradigm and we have added it as a fourth architecture in Appendix E to increase the scope of architectures we study.
>
> Our DeepONet implementation follows the standard branch-truck decomposition with slight changes adapting it for the inverse problem setting. The branch network, which processes the PDE solution at collocation points, takes in the full spatiotemporal solution and is implemented as a ResNet, primarily for computational efficiency when handling the full grid. The trunk network for encoding sensor locations is implemented as a pointwise MLP with residual connections and GeLU activations. Since most of our inverse problems predict a scalar PDE parameter rather than a spatio-temporal field, we apply mean pooling over the pointwise branch-trunk inner produce to produce the final scalar output.
>
> Across systems, we find that DeepONet performs similarly to ResNet (see Appendix Figure 20). We attribute part of this to the reuse of ResNet as the branch network in the DeepONet setup. A full design exploration of DeepONets is left as future work.
>
> ### 5. NLS Metric
> > A few words to better justify the NLS metric would likely strengthen the text.
>
> We have added some explanation/citations relating the NLS metric to well-established approaches in the neural scaling law and machine learning force field literature:
>
> [1] Bahri, Yasaman, et al. "Explaining neural scaling laws." *Proceedings of the National Academy of Sciences* 121.27 (2024): e2311878121.
>
> [2] Hestness, Joel, et al. "Deep learning scaling is predictable, empirically." *arXiv preprint arXiv:1712.00409* (2017).
>
> [3] Batzner, Simon, et al. "E (3)-equivariant graph neural networks for data-efficient and accurate interatomic potentials." *Nature communications* 13.1 (2022): 2453.

---

> > ### Author Response · Authors · 2026-04-17
> > **Follow up**
> >
> > Hello. We would like to follow up regarding our response to your review. Please let us know if our response addressed your concerns. Thank you.

---

> > > ### Comment · Reviewer_Xivm · 2026-04-21
> > >
> > > My apologies for the delay. Thank you for the added details & simulations. I already liked the first version of the manuscript, and I like the revised one even better.

---

### Comment · Action_Editor_6qWB · 2026-03-13
**Response to reviews**

Dear authors:

Thank you for your submission. Now that all three reviews are public and the discussion phase has started, please note that you have up to two weeks from the start of the discussion phase to post your author response. We encourage you to address all reviewer comments and questions as clearly and concretely as possible.

---

> ### Author Response · Authors · 2026-04-02
> **Response to Reviewers and Revision Posted**
>
> We sincerely thank the Action Editor and the reviewers for their helpful feedback. Their comments improved the quality and clarity of the manuscript. We have revised the paper accordingly and included additional experiments in the supplementary appendix. Below, we provide detailed responses to each comment.

---

### Decision · Action_Editor_6qWB · 2026-04-21

**Recommendation:** Accept as is

**Audience:**

Yes

**Audience Explanation:**

This work will attract researchers in AI4Science.

**Claims And Evidence:**

Yes

**Claims Explanation:**

This paper introduces PDEInvBench, a public benchmark for PDE inverse problems. The benchmark cover several representative systems such as 2D reaction–diffusion, 2D Navier–Stokes, 1D KdV, and 2D Darcy flow. Through a systematic study across training paradigms, model families and scaling axes, the authors provide some useful findings for practitioners. While reviewers note limitations such as noise-free observations and future extensions to partial observations/irregular meshes/3D, the dataset, protocol, and analyses fill an important gap in scientific ML benchmarking for inverse problems.